

# Optimizing UV index determination from broadband irradiances

Keith. A. Tereszchuk[1], Yves. J. Rochon[1], Chris. A. McLinden[1], and Paul. A. Vaillancourt[2]

[1]Air Quality Research Division, Environment and Climate Change Canada, Toronto, Ontario, Canada
[2]Meteorological Research Division, Environment and Climate Change Canada, Dorval, Quebec, Canada

*Correspondence to:* Keith. A. Tereszchuk
(keith.tereszchuk@canada.ca)

**Abstract.** Amidst mounting concerns about the depletion of stratospheric ozone ($O_3$), and for subsequent increases in the surface irradiances of ultraviolet (UV) light and its effects on human health, a daily UV forecast program was launched by Environment Canada in 1993. The program serves to monitor harmful surface UV radiation and provide this information to the Canadian public through the UV index, a scale which reports the relative intensity of the Sun's UV radiation at the Earth's

surface, and the corresponding protection actions to be taken. The UV index was accepted as a standard method of reporting surface UV irradiances by the World Meteorological Organization (WMO) and World Health Organization (WHO) in 1994.

A study was undertaken to improve upon the prognosticative capability of Environment and Climate Change Canada's (ECCC) UV index forecast model. An aspect of that work, and the topic of this communication, was to investigate the use of the four UV broadband surface irradiance fields generated by ECCC's Global Environmental Multi-scale (GEM) numerical

prediction model to determine the UV index.

The basis of the investigation involves the creation of a suite of routines which employ high spectral resolution radiative transfer code developed to calculate UV index fields from GEM forecasts. These routines employ a modified version of the Cloud-J v7.4 radiative transfer model, which integrates GEM output to produce high spectral resolution surface irradiance fields. The output generated using the high-resolution radiative transfer code served to verify and calibrate GEM broadband

surface irradiances under clear-sky conditions and their use in providing the UV index. A subsequent comparison of irradiances and UV index under cloudy conditions was also performed.

Linear correlation agreement of surface irradiances from the two models for each of the two higher UV bands covering 310-330 nm and 330-400 nm is typically greater than 95% for clear-sky conditions with associated root mean square relative errors of 5.5% and 3.8%. On the other hand, underestimations of clear-sky GEM irradiances were found on the order of ∼30-

50% for the 294-310 nm band and by a factor of ∼30 for the 280-294 nm band. This underestimation can be significant for UV index determination but would not impact weather forecasting. Corresponding empirical adjustments were applied to the broadband irradiances now giving a correlation coefficient of unity. From these, a least-squares fitting was derived for the calculation of the UV index. The resultant differences in UV indices from the high spectral resolution irradiances and the resultant GEM broadband irradiances are typically within 0.2 with a root mean square relative error in the scatter of ∼5.5% for

clear-sky conditions. Similar results are reproduced under cloudy conditions with light to moderate clouds, having a relative error comparable to the clear-sky counterpart; under strong attenuation due to clouds, a substantial increase in the root mean square relative error of up to 30% is observed due to differing cloud radiative transfer models.





# 1   Introduction

Throughout the late 1980s and early 1990s, extensive atmospheric studies in the polar regions of the planet revealed that stratospheric ozone ($O_3$) concentrations were being depleted due to a variety of $O_3$ destroying catalytic cycles driven by photochemical reactions liberating chlorine (Cl) and bromine (Br) atoms from chlorofluorocarbon (CFC) and hydrofluorocarbon

(HCFC) molecules emitted into the atmosphere as airborne anthropogenic pollutants (Rowland , 1996).

Ozone is an important atmospheric absorber of energetic, short-wavelength, radiation emitted by the Sun. Most critically, $O_3$ is the primary absorber of ultraviolet (UV) radiation, which has wide-ranging implications on the health of the biosphere; both on a molecular level with the potential of damaging the cellular DNA of individual organisms (Ravanat et al. , 2001), to the destabilization of entire biogeochemical cycles within a biome (Zepp et al. , 1998).

UV radiation is categorized into three broadband regions which are defined as: UV-A (315-400 nm), UV-B (280-315 nm) and UV-C (100-280 nm). Molecular species in the Earth's atmosphere absorb very little of the longer wavelength UV-A radiation, as it reaches the surface with a minor net difference (mainly due to scattering) in the radiative flux from the top of the atmosphere. UV-B radiation is partially transmitted through the atmosphere and is primarily absorbed by $O_3$ (Huggins/Hartley band system). The Huggins/Hartley system ($\sim$200-360 nm) of $O_3$ and the Hopfield/Schumann?Runge system ($\sim$70-200 nm) of molecular

oxygen $O_2$ serve to absorb all UV-C radiation, which is impeded from reaching the top of the troposphere. This absorption occurs primarily in the ozone layer, a thin band of $O_3$ contained within the stratosphere where the peak molecular number density of $O_3$ is located $\sim$20-30 km above sea level. Figure  1 demonstrates how the absorption by ozone increases rapidly with decreasing wavelength in the UV-B region, causing surface irradiances to fall off sharply with decreasing wavelength.

At progressively shorter wavelengths of UV light, increasingly energetic photons become subsequently more and more

damaging to biological species, including humans. Studies were conducted as early as the 1930s to quantify the damage done to human skin by UV radiation. It had been well known for quite some time that UV-A and UV-B radiation is harmful to unicellular organisms, the surface cells of plants and animals, and to the health of the more sensitive population. Colblentz and Stair  (1934) sought to obtain measurements of the spectral erythemic reaction (reddening) of untanned human skin exposed to UV light. In essence, this was one of the first recordings of a UV erythemal action spectrum, where an action spectrum for

a particular biological effect expresses the effectiveness of radiation at each wavelength as a fraction of the effectiveness at a certain standard wavelength. In this case, the tolerance of human skin to ultraviolet radiation. Today, research has revealed that humans are susceptible to much more than sunburns when exposed to UV rays. Prolonged exposure can lead to the premature aging of the skin, suppression of the immune system, eye damage including the development of corneal photokeratitis and





cataracts, and skin cancer (melanoma). The contemporary action spectrum adopted by most international organizations is the CIE (Commission Internationale de l'Éclairage, International Commission on Illumination) action spectrum using the method outlined by McKinlay and Diffey (1987); CIE Technical Report (2014). The piecewise function in Eq. (1), which mathematically represents the McKinlay-Diffey erythemal action spectrum, is also detailed in Fig. 1.

$$
EAS(\lambda) = \begin{cases} 1 & 250 < \lambda \leq 298 \\ 10^{0.094(298-\lambda)} & 298 < \lambda \leq 328 \\ 10^{0.015(139-\lambda)} & 328 < \lambda \leq 400 \\ 0 & 400 < \lambda \end{cases} \quad [\lambda \text{ nm}] \tag{1}
$$

The UV index was developed as an erythemally weighted representation of the total surface flux of UV radiation in the biologically active range of 280-400 nm (CIE Technical Report , 2014; Fioletov et al. , 1997; Allaart et al. , 2004; Fioletov et al. , 2010; Moshammer et al. , 2016); the range below ~280-290 nm can be excluded as its contribution is negligible. It was conceived to produce a simplified scale which reports the relative strength of the Sun's UV radiation, and to inform the public of the Sun protection actions that should be taken as a precaution if they are to be exposed to the Sun's rays for extended periods of time.

To determine the UV index from high spectral resolution irradiances, an effective spectral curve is calculated from the product of the erythemal action spectrum and the surface irradiance (Fig. 2). This effective curve, the weighted UV irradiance, is then integrated over the spectral range representing the UV-A and UV-B (280-400 nm) to produce the UV index (see Eq. (2)). A scaling factor of $(25\text{mW/m}^2)^{-1}$ is implemented to provide a convenient set of integer values, normally ranging from 0 to 11. In extreme cases, values of >11 can be reached and are typically recorded in the tropics where the solar zenith angle is at a minimum. Extreme values are also recorded at high elevations where the atmospheric optical path is shortened, resulting in a reduced attenuation of actinic fluxes and consequently producing increased surface irradiances.

$$
UVI = \frac{1}{25 \frac{\text{mW}}{\text{m}^2}} \int_{280\,\text{nm}}^{400\,\text{nm}} I(\lambda) \cdot EAS(\lambda) \mathrm{d}\lambda \tag{2}
$$

Following concerns arising in the late 1980s from the escalating loss of stratospheric $O_3$ due to CFCs, and the subsequent increases in the surface irradiances of UV radiation (Crutzen , 1992), Environment and Climate Change Canada began providing daily UV index forecasts as of 1992 (Burrows et al. , 1994). Since its inception in 1992, the UV index has been adopted worldwide as standard indicator to characterize solar UV intensity at the Earth's surface (Fioletov et al. , 2010) and serves to inform the public about the strength of the Sun's UV radiation and the adequate sun protection actions recommended to avoid excessive exposure to UV radiation.

At present, the UV index determination for the ECCC forecast system relies on a statistically derived weather-based computation of the total column ozone field, adjustments using total column measurements of the Canadian Brewer network and empirical conversions to the UV Index accounting for the solar zenith angle, cloud conditions, surface altitude and snow cover. A recently undertaken study toward improving the UV index forecast system makes direct use of ozone data assimilation, ozone model forecasts, and model UV irradiance forecasts for both clear-sky and cloudy conditions as done in some capacity



at other forecast centers (e.g., NCEP/NOAA, KNMI, and ECMWF). A summary of UV index forecasting practices conducted by various governmental organizations worldwide were compiled by Long (2003).

This current study is part of a multi-faceted project which seeks to include having a UV index forecasting package more tightly integrated into the current weather (and air quality) forecasting system, and increasing the array of UV index products available from ECCC to Canadians, such as daytime variation, longer forecasts, and continental and regional maps. The ECCC Global Environmental Multi-scale (GEM) numerical weather prediction model provides four broadband irradiances shown in Fig. 2 covering the UV spectrum in the range of 280-400 nm, which can be calculated using three-dimensional prognostic ozone fields. The work presented in this communication consists of investigating and optimizing the calculation of the UV index from these broadband irradiances, with focus on clear-sky conditions, for minimizing computational cost and processing time. This is done through comparisons of the UV index and broadbands irradiances produced from GEM to those calculated using the Cloud-J radiative transfer model, which has been adapted to provide high resolution irradiance spectra at the Earth's surface.

The following subsections provide some background on the GEM-based weather forecast system, the Cloud-J radiative transfer model, and their products. Section 2 describes the general methodology and the related fitting approaches applied in Sect. 3 to investigate and optimize the calculation of the UV index from the broadband irradiances through the use of high-resolution spectral irradiance simulations for clear-sky conditions. While a specific optimization under cloudy conditions is not performed due to differing cloud radiative transfer models, comparisons for both clear and cloudy conditions are presented and fully discussed in Sect. 3. Conclusions are provided in Sect. 4.

## 1.1 GEM with LINOZ

The irradiance fields calculated by GEM are based on the cccmarad radiative transfer scheme which uses a correlated-k distribution method for gaseous transmission detailed by Li and Barker (2005) and von Salzen et al. (2013). The Li and Barker (2005) radiation scheme has four wavenumber intervals for the shortwave and nine intervals for the longwave. The visible and UV portion of the shortwave is further subdivided into 9 subbands. The four subbands of relevance to the calculation of the UV index cover the following spectral ranges: 280-294 nm, 294-310 nm, 310-330 nm, 330-400 nm. The irradiances of the subbands, i.e., the broadband irradiances, consist of direct and diffuse components which are available in addition to their sum. Also differentiated, are the clear sky and the total (clear+cloudy) sky analog for the total surface irradiance, as well as differentiation for each of the individual direct and diffuse components of the subbands.

The GEM dynamical core is described in Girard et al. (2014), while basic descriptions of the physical parameterizations or detailed references can be found in Zadra et al. (2014a, b). Model runs were performed using a 7.5 minute time step for a uniform 1024x800 longitude-latitude grid ($0.352°\times0.225°$) and a Charney-Phillips vertically staggered grid with 80 thermodynamic levels extending from the near-surface (at $\eta = 1$) to $\sim0.1$ mbar ($\eta \approx 0.0001$). The analyses, serving as initial conditions for providing the forecasts used in this study, are a composite of the already available ECCC weather analysis and separately generated ozone analyses. The GEM forecast products used as input for the simulations performed with Cloud-J are detailed in Sect. 2.1.





Prognostic ozone is solved with a linearized photochemistry scheme called LINOZ (McLinden et al. , 2000), which was implemented on-line within the GEM NWP model (de Grandpré et al. , 2016). For this work, the ozone analyses stem from assimilation of total column ozone data obtained from the National Environmental Satellite, Data, and Information Service (NESDIS/NOAA) for the Global Ozone Monitoring Experiment-2 (GOME-2) instruments of the MetOp-A and MetOp-B

satellites (Callies et al. , 2000; Munro et al. , 2006) . Assimilations were performed with the incremental three-dimensional variational approach with the first guess at appropriate time (FGAT; Fisher and Andersson (2001)) using elements of the system described in Charron et al. (2012), and the references therein, adapted for chemical data assimilation.

For the treatment of cloud, GEM employs a prognostic total cloud water variable with a bulk-microphysics scheme for non-convective clouds. The radiative transfer impact from clouds is primarily dictated by the liquid and ice water mixing ratios

(LWCR and IWCR) and cloud fraction (CLDR). Fractional cloudiness is based on a relative humidity threshold, which varies in the vertical. Individual cloud layers are assumed to overlap in the vertical using a maximum random cloud overlap (Sundqvist et al. , 1989; Paquin-Ricard et al. , 2010).

The GEM model currently does not assimilate aerosol measurement data. The radiative effects associated with background aerosols are based on a climatology produced by Toon and Pollack (1976). This climatology specifies maximum aerosol

loading at the equator and a decrease toward the poles, with different values for continents and oceans. These distributions also include a latitudinal gradient. Aerosols are assumed only to affect the solar absorption properties of the clear-sky atmosphere (Markovic et al. , 2008).

## 1.2 Cloud-J

Cloud-J, a recent release of the Fast-J program (Wild et al. , 2000; Bian and Prather , 2002), is a multi-scattering, eight-stream,

radiative transfer model for solar radiation (Prather , 2015) developed for integration into three dimensional chemical transport models to calculate photolysis rates ($J$ values) in the atmosphere. The version of the program used for this work is Cloud-J v7.4. The program is developed and maintained by M. Prather in the Department of Earth System Science at the University of California, Irvine (http://www.ess.uci.edu/group/prather/scholar_software/cloud-j).

To calculate photolysis rates, the standard Cloud-J code uses 18 interpolated wavelength bins covering a spectral range

of 187-599 nm. The integrated radiative transfer model uses a plane parallel atmosphere assumption and a full scattering phase function. Rayleigh and isotropic scattering are taken into consideration. Numerous cloud types and aerosol species of varying sizes are accounted for in the calculations by making use of look-up tables containing the scattering functions for water droplet size, ice crystals of various phases, dust, absorbing soot (black carbon), stratospheric sulfates (background and volcanic) and water haze at 0.1 $\mu$m and 0.4 $\mu$m. Optical depth properties include extinction optical depth, single scatter albedo,

and a scattering phase function.

Cloud-J provides numerous options for the treatment of clouds in its radiative transfer calculations. Option 1 is the calculation for clear-skies conditions. Option 2&3 are variations of the direct use of the cloud water content, which employs cloud fraction and separate liquid and ice water paths. The remaining five options (4-8) employ different variations in the correlated, overlapping cloud scheme. The approach seeks to represent the fractional cloud cover in the model layers through the calcula-





tion of numerous independent cloud atmospheres (ICAs), where each ICA would be either 100% cloudy or clear in each cell of the cloud model layer. This fractional cloud-overlap model serves to determine the layer structure, weighting, and number of ICAs that best represent the actual cloud distribution in the model layers.

## 2 Methodology

Given the availability of realistic three-dimensional prognostic ozone to the GEM numerical weather prediction model through the LINOZ linearized ozone model and ozone data assimilation, it was proposed to make direct use of the four GEM model UV broadband irradiances at the Earth's surface to calculate the UV index. The Cloud-J radiative transfer model was adapted to provide high spectral resolution surface irradiances in the UV, 280-400 nm. The high resolution output is used to evaluate the GEM broadband irradiances for clear-sky conditions and to optimize the determination of the UV index using these coarse

resolution spectral broadbands. A comparison of results from the two models under cloudy conditions is also performed in Sect. 3.

To perform the optimization of the GEM broadbands, the desired output from Cloud-J is two-fold:

1. Sets of Cloud-J broadband irradiances generated by integrating portions of the high resolution irradiance spectra to produce simulated versions of the four GEM UV broadbands covering 280-294 nm, 294-310 nm, 310-330 nm and

15 330-400 nm.

2. A global UV index field produced by integrating the erythemally weighted high-resolution irradiance spectra over the 280-400 nm spectral range, Eq. (2).

Simulated broadbands are generated for comparison with the GEM broadband irradiances and, as needed, used to create sets of scaling functions to calibrate the GEM values to the Cloud-J output. The scaled GEM broadbands are then weighted

accordingly such that the global UV index field produced using the GEM broadbands emulates the high resolution UV index field calculated from Cloud-J. Two different approaches were implemented to calculate the UV index from the resultant GEM broadband surface irradiances. A least-squares fitting was employed in both cases to optimize the weighting under clear-sky conditions using the UV index field produced from the high-resolution Cloud-J spectra as a reference.

The following subsections briefly describe the application of GEM products and the Cloud-J model to ultimately evaluate

and optimize the UV index determination from the broadband irradiances.

### 2.1 Calculation of high spectral resolution irradiances

Originally designed to calculate tropospheric/stratospheric photolysis rates in 3D global models, the Cloud-J program was adapted to input three-dimensional fields from the GEM model and output direct and diffuse, high spectral resolution, surface irradiances instead of mean photolytic intensities. The resultant surface spectral irradiances are, in turn, used to calculate UV

index fields.



To produce the high spectral resolution output for UV index calculations, the number of wavelength bins was increased to 241 with 0.5 nm intervals over the 280-400 nm spectral range. Having augmented the number of wavelength bins to perform the high resolution calculations, additional spectroscopic data was required for integration into Cloud-J. These spectral parameters were interpolated onto a 0.5 nm resolution grid and reformatted for reading into the program along with the GEM model

forecasts. The spectral data incorporated into Cloud-J include:

- A set of UV-Visible temperature/pressure absorption cross-sections for $O_3$ obtained from the GEISA spectroscopic database (Jacquinet-Husson et al. , 2008).

- An Earth surface reflectance climatology from five years (2005-2009) of OMI data (Kleipool et al. , 2008). Surface reflectivities are provided as monthly averages for 23 wavelength channels, 328-499 nm range, on a 0.5°x0.5°grid.

- A high-resolution, top of atmosphere (TOA), solar flux spectrum between 250 and 550 nm (Dobber et al. , 2008). Provided by Q. Kleipool of the Royal Netherlands Meteorological Institute, the reference spectrum was created to calibrate/validate the Ozone Monitoring Instrument (OMI).

- Rayleigh scattering parameters calculated using the methodology detailed in a publication by Chance and Spurr (1997).

The $O_3$ cross-sections obtained from the GEISA database (http://www.pole-ether.fr/ether/pubipsl/GEISA/geisa_crossUV_

frame_2011_uk.jsp) were recorded by Voight et al. (2001) on a Bruker IFS 120HR Fourier-transform spectrometer at a spectral resolution of 5.0 cm$^{-1}$). The measurements were performed as a follow up to the cross-sectional data initially recorded by Burrows et al. (1999) on the GOME-FM instrument. The new data sets recorded by Voight et al. (2001) offer precise reference spectra where the spectral accuracy of the data is better than 0.1 cm$^{-1}$ ($\sim$0.5 pm at 230 nm and $\sim$7.2 pm at 850 nm), which was validated by recording visible absorption spectra of gaseous diatomic iodide $I_2$ in a reference cell using the same experimental

set-up. The agreement between observed and modelled data was determined to be 1% and better within the 255-310 nm region. Sets of $O_3$ absorption spectra were recorded using total pressures of 100 mbar and 1000 mbar at five different temperatures ranging from 203-293 K. The spectra in the UV range at 100 and 1000 mbar are nearly identical with larger differences at higher wavenumbers. Three $O_3$ absorption spectra from this data set were used for incorporation into Cloud-J (1000 mbar at 293 K, 100 mbar at 246 K and 223 K). The selection of the three spectra was based on consideration of the typical temperature

distribution as a function of pressure.

In addition to the $O_3$ temperature cross-sections, the $O^1D$ quantum yields associated with ozone photolysis were also required by the Cloud-J radiative transfer model. Values for the quantum yields were calculated using the prescribed method outlined by Matsumi et al. (2002) for the same three temperatures associated with the GEISA $O_3$ cross-sections.

The albedo data was interpolated from its native grid onto the 1024x800 GEM global grid. A linear interpolation was then

performed on the data from the 23 re-gridded wavelength channels to obtain the intermediate albedo global fields corresponding to 0.5 nm intervals over the 328-400 nm spectral range to be subsequently used in the high resolution irradiance calculations. Albedo values for the bins corresponding to the missing wavelength range of 280-328 nm were obtained by linearly interpolating the data between the 328 nm OMI channel and the UV-B values published by Chadyšien and Girgždys (2008). According





to the experimental data reported in Table 2 of Chadyšien and Girgždys (2008), snow/ice is the primary reflector of UV-A and UV-B radiation where surface reflectivity for these spectral regions are 94% and 88% respectively, representing a drop in reflectivity of 6.38% in the shorter wavelength region. To emulate the experimental data, the reflectivities for the 328 nm OMI channel were linearly reduced by 6.38% over the 280-328 nm spectral range.

The OMI solar reference spectrum produced by Dobber et al. (2008) was used to provide the TOA solar flux values required for the high-resolution irradiance calculations performed by Cloud-J. Currently, there are no high resolution solar spectra that cover the UV-A and UV-B wavelength ranges. Most UV/Vis TOA spectra are pieced together from different sources in order to provide a continuous, unbroken spectrum. The OMI reference spectrum was created to validate the radiometric calibration of OMI measurements and to monitor potential optical degradation of the instrument. Also a combined spectrum, it was produced

by employing the approach used by Chance and Spurr (1997). It merges the balloon spectrum of Hall and Anderson (1991), which covers a shortwave UV region between 200 and 310 nm, with a ground-based spectrum obtained from the McMath-Pierce solar telescope at Kitt Peak National Observatory (Kurucz et al. , 1984). The broadband Kitt Peak spectrum covers a spectral range of 296 and 1200 nm. The final derived spectrum is at 0.01 nm sampling and at 0.025 nm resolution.

This spectrum was chosen for use in this work, from amongst others, because the OMI composite spectrum uses high

resolution (0.01 nm) UV measurements made in the stratosphere from a balloon at ∼40 km in altitude (Hall and Anderson , 1991) to avoid affects of the strong atmospheric absorption below 300 nm Dobber et al. (2008). The solar reference spectrum produced by Thuillier et al. (2003) was also considered since it is composed from measurements made from the SOLSPEC and SOSP satellite instruments (Thuillier et al. , 1998, 2003) with a resolution of 1 nm. With both spectra being similar, the former was selected due to its higher spectral resolution even though the resolution of the latter is only a factor of two coarser

than our simulation resolution. A moving boxcar averaging window covering ±0.25 nm about sampling points at intervals of 0.5 nm was applied to the OMI composite spectrum to generate the simulation spectrum.

Consideration was also given to high resolution spectra based on accurate models of the Sun using the Kurucz et al. (1984) spectrum, such as those by Chance and Spurr (1997); Chance and Kurucz (2010), which provide excellent spectral range, sampling, and resolution. These spectra unfortunately neglect optimization in the UV-B region for radiometric accuracy.

The SAO96 and re-calibrated SAO96 (SAO2010) reference spectra described by Chance and Kurucz (2010) both utilize the original Kurucz et al. (1984) Kitt Peak spectrum for the UV-B, where $O_3$ structure was not fully removed. Chance and Spurr (1997) reported that efforts were focused on intensity calibration of the wavelength range where most application to satellite measurements are performed. Intensities for portions of the spectrum shortward of 305 nm may be in substantial disagreement, by as much as 20%, with both the Dobber et al. (2008) and Thuillier et al. (2003). These spectra were deemed unsuitable for

use in the calculation of the UV Index.

It should be noted that the solar spectrum used in this work is representative of a yearly average value of the Earth's TOA flux. Changes in the Earth-Sun distance and associated solar fluxes during the Earth's annual cycle are taken into account and are corrected for by Cloud-J in the high-resolution simulations.

The input atmospheric conditions provided to Cloud-J for this study consist of a set of 6 hour forecasts from the GEM model

output for the dates of August 23-29, 2015, at 18h00 UTC with daytime over North America. The GEM fields provided as





input are surface pressure, and the three-dimensional fields of temperature, pressure (derived from the vertical coordinate and surface pressure), ozone, specific humidity (converted to relative humidity), liquid and ice water mixing ratios (LWCR and IWCR), and cloud fraction (CLDR). For the all-sky conditions, the parameters LWCR, IWCR and CLDR determine the liquid and ice water partial column amounts (in g/m$^2$) of each model layer in the presence of clouds.

Cloud-J was run individually for each day during the period of August 23-29 to produce irradiance fields representing the direct, diffuse and total surface flux under both clear-sky and all-sky conditions. Weekly (7-day) averages of the direct, diffuse and total spectral irradiances served as reference spectra in the least-squares minimization for evaluation and adjustments of GEM broadband irradiances, with individual forecast values used in the scatter plot comparisons. The UV indices produced with Eq. (2) from weekly averages of total spectral irradiances served as reference in optimizing of UV index estimation models
based broadband irradiances. The UV index field from the clear-sky weekly averages is shown in Fig. 3.

## 2.2    Comparison to ground-based clear-sky irradiances

In addition to measuring total column ozone, Brewer spectrophotometers provide ground-based measurements of the UV spectrum in the range 290-325 nm with an effective resolution of about 0.55 nm and a sampling interval of 0.5 nm. The data processing scheme used to generate spectral irradiances at each 0.5 nm interval, which includes calibration and corrections
for various factors, is described in the work detailed by Kerr  (2010). A sample inter-comparison of three Brewer instruments by Thompson et al.  (1997) (see Kerr  (2010) for other inter-comparison sources) showed relative overall differences between instruments within 6% with an average of 3% for wavelengths longer than 300 nm; uncertainties are larger at shorter wavelengths.

Cloud-J clear-sky surface UV irradiances were compared to Brewer spectra obtained from six different measurement stations
belonging to ECCC's ozone monitoring network and identified to be under clear-sky to optically thin cloud conditions. The applied TOA solar spectrum used here for the Cloud-J simulations, as well as for optimizing use of the GEM broadband irradiances in UV index calculations, has the same sampling interval of 0.5 nm as the Brewer measurements and a similar effective resolution of 0.5 nm. For the latter, a boxcar averaging window was applied instead of the trapezoid-shaped Brewer slit function. Figure 4 depicts 5-day averages of Brewer measurements taken at 18h00 UTC on random days in the months of
July and August of 2015 and the equivalent counterpart irradiance spectra calculated from Cloud-J. The locations were chosen to provide *in-situ* measurements for different solar zenith angles in addition to varying geographic locations to evaluate the level of agreement between the Cloud-J model application and the Brewer spectra. The Cloud-J derived spectral irradiance curves largely follow those recorded by the Brewers. The differences between the sets of curves give an overall root mean square relative error between the Cloud-J and Brewer spectra of ∼15%.

The overall differences in the Cloud-J and Brewer data were also quantified by integrating the spectra of each of the six stations to produce sets of broadbands covering the 295-310 nm and 310-325 nm regions, the 310 nm node denoting the point transition between the similarly corresponding GEM irradiance broadbands. The resultant mean percent differences of the Cloud-J broadband values compared to the ground-based measurements for the 295-310 nm and 310-330 nm bands are 0.9%±4.0% and 7.5%±1.4%, respectively. While not clearly evident from the figure for the lower band, the largest band





differences were obtained for Resolute at 5.3% and 9.9%, respectively. Different sources possibly contributing to the variations observed between the model and measurement for each station include: disparities in clear-sky to light cloud conditions, surface reflectivities, air pollution, column ozone, and in the actual locations and heights between the Brewer stations and the nearest corresponding model grid points used to represent these locations. Differences in height above sea level between the model

grid points and station locations are under 30 m except for Saturna (Fig. 4e) at 26 m versus 202 m and Eureka (Fig. 4c) at 159 m versus 9 m. For example, the lower grid point height for Saturna might be contributing to lower irradiances in the 295-310 nm range relative to most stations when comparing to the Brewer data.

Average differences in total column ozone between the GEM model ozone fields provided to Cloud-J simulations and the Brewer measurements for the sample data set of the figure in the range of 2.8 to 4.4% for the four non-Arctic stations and 0.5

and 0.4% for the two Arctic stations of Eureka (Fig. 4c) and Resolute(Fig. 4d). It was determined that the GOME-2 column ozone data used in the assimilation to generate the model forecasts were similarly biased relative to Brewers for that period; satellite data bias can be reduced through corrections such as in van der A et al. (2015). Correcting for the ozone differences would increase the Cloud-J irradiances, and thus the differences with Brewers, by at least 3-5% in the lower band for the non-Arctic stations and much less so for the Arctic stations, correspondingly increasing the positive differences with the Brewer

spectra. The higher band would be less affected as absorption from ozone is comparatively weaker for the upper wavelengths. This would bring the 295-310 nm band irradiance differences in percentage closer to the 310-325 nm differences.

Further analysis of the data sets depicted in Fig. 4 reveal that the ratio of the 310-325 nm to 295-310 nm bands is 25 for the two Arctic stations and 15 to 17 for the four non-Arctic stations. This illustrates the relative increase of irradiances above versus below 310 nm for increasing solar zenith angles associated to the stronger increased atmospheric attenuation by ozone

in the lower band. As the contribution of the 295-310 nm band to large UV index values (low solar zenith angles) is more dominant, the impact of differences above 310 nm would become more visible on low UV index values (high solar zenith angles). Implications of the differing sizes of differences between the 295-310nm and 310-330 nm bands and in model ozone forecasts are examined further in Sect. 3.1.3.

The source of the usually larger Cloud-J irradiances is not known. This is larger than the 3% uncertainty from the three

Brewers inter-comparison by Thompson et al. (1997) but within the spread of mean differences in Bais et al. (2001) over the different Brewers and instruments of other types from the SUSPEN inter-comparison for wavelengths above 300-305 nm. The latter does not however address the overall consistency of the differences of the Cloud-J results over the six Brewers. The solar irradiance changes due to the changing orbital earth-sun distance are reflected in the simulations. The sun itself displays cyclical short-term (solar rotation) and long-term (solar cycle) solar spectrum irradiance variability. In the UV index spectral

range, these changes are within roughly 0.2% and 0.6-1.5% based on measurements over the recent decades (Yeo et al. , 2015; Marchenko et al. , 2016; Mathes et al. , 2017); the total irradiance has a weaker solar cycle change of 0.1%. These are too small to account for the differences seen in the 310-325 nm band. Taking the Brewer spectra as reference, the above would suggest a scaling factor adjustment of the Cloud-J irradiances roughly of size 0.93. No such scaling is applied in this paper.





### 2.3 Estimation of the UV index from GEM broadband irradiances

Two UV index estimation approaches using the four broadband irradiances were considered. One consists of linear fitting directly to three of the four UV broadband irradiances, i.e.

$$UVI = w_1 I_{280\text{-}294} + w_2 I_{294\text{-}310} + w_3 I_{310\text{-}330} + w_4 I_{330\text{-}400} \tag{3}$$

with $I_{\Delta\lambda}$ in W/m² and fit coefficients $w_i$. With this equation, the contribution from the lowest band can be neglected unless the total column ozone is less than roughly 210 DU to contribute at least 0.1 units to the UV index. Its coefficient value $w_1$ is analytically derived to be 40 m²/W since the erythemal action spectrum is constant over the spectral range of the lowest band.

The other approach involves applying the integral of Eq. (2) to piecewise interpolated spectra. Both fits are intended to have the UV index values derived from the broadband irradiances be consistent with the values obtained from the integrated high
resolution effective spectra. UV index values larger than 3 are used in the minimization to focus the weighting on regions of moderate to high UV index values. The fitting over points with UV index values larger than 3 does not exclude points and regions with isolated outlier differences and includes both land, water, and snow/ice surfaces. Minimization was performed using an amoeba downhill simplex method employing a least-squares fitting of the UV index fields from the scaled GEM broadband irradiances to those from the high resolution spectra produced by Cloud-J.

For the integral approach, the available irradiances in W/m² over the four UV spectral broadbands must be transformed to spectral irradiances for multiplication to the erythemal function prior to spectral integration. The approximate conversion to spectral irradiances is done as follows:

1.  The band irradiances are divided by the band widths to generate average spectral irradiances.

2.  Each of the resulting average spectral irradiances in W/(m²/nm) is associated with a particular reference spectral position
to be determined through fitting.

3.  Logarithmic first or second order Lagrange interpolation is applied over each piecewise spectral integration interval without forcing agreement at the band interfaces.

The selected order of the logarithmic interpolations and initial estimates of the spectral reference positions were chosen through trial and error. The optimized spectral positions are determined through least-squares fitting to the UV index values
calculated from the Cloud-J high spectral resolution irradiances.

Interpolations and weighted integrations are performed over four segments covering the ranges 294-298, 298-310, 310-328 and 328-400 nm. The irradiance for 280-294 nm is simply added to the sum of the integrations over the above four ranges as the erythemal function is constant with a value of unity over that spectral range; its contribution over this integration segment could alternatively be omitted as it is negligible. Determination of a reference spectral irradiance for this first band in step 2
above is still done to provide a required interpolation node for the other integration segments. The specification of the segments is dictated by the band widths and the two positions, 298 nm and 328 nm, of the slope changes in the erythemal function. The



applied interpolations are of second order for the ranges 294-298 and 310-328 nm and are linear in the other ranges. The simple interpolations do not strictly preserve the original broadband irradiance values nor accurately replicate high resolution spectra since the main interest is the fast computation of good estimates of the resultant integral value. The integrations for the last three segments are done using Simpson?s rule with two subintervals (5 interpolation nodes) and that for 294-298 nm is done with one interval (3 interpolation nodes).

## 3   Results

### 3.1   Clear-sky conditions

#### 3.1.1   Broadband irradiances

The comparisons made between GEM and Cloud-J broadband irradiances for clear-sky conditions shows a fairly good agreement in the 310-330 nm and 330-400 nm bands. For these bands, the linear correlation agreement between the two models is typically greater than 95% with associated root mean square relative errors of 5.5% and 3.8% for midday values. On the other hand, underestimations of GEM irradiances were found in the order of $\sim$30-50% for 294-310 nm band and by a factor of $\sim$30 for the 280-294 nm band as shown in Fig. 5b and Fig. 5a, respectively. It was subsequently identified that the bulk of the differences for the two lower bands, especially the disparity in curvatures in bands 1 and 2 of Fig. 5, stems from differences in equivalent broadband absorption cross-sections if not also TOA solar fluxes. This is further supported by the significantly improved agreement demonstrated in Fig. 6 where the cross-sections of the correlated-$k$ approach cited in Table 6 of Li and Barker  (2005) and the solar broadband top of the atmosphere (TOA) fluxes employed by the GEM model were instead applied in the Cloud-J calculations. It should be noted that the band solar fluxes used in GEM differ by approximately 0.02% to 0.15% from the UV sub-band solar fluxes reported in Table 6 of Li and Barker  (2005).

A direct comparison was made between the GEM TOA solar fluxes and the broadband averages that were calculated from Cloud-J using the data obtained from Dobber et al.  (2008). There are significant differences in the two short-wavelength broadbands with the band values calculated from the Dobber et al.  (2008) fluxes being smaller than the GEM fluxes by 35% and 15% for the 280-294 and 294-310 nm bands, respectively; values for the higher bands are only 3% smaller and 2% larger, respectively. These differences would favour an underestimation of the Cloud-J irradiances relative to GEM at the shorter wavelengths in the absence of differences in cross-sections, which is opposite to the results in Fig. 5. A comparison to the band averages derived from the solar flux spectrum of Chance and Kurucz  (2010) gives smaller differences of -12%, 2%, -1% and 3.5% relative the GEM values.

The spectrally, uniformly-weighted average cross-sections from the GEISA dataset which represent the four UV broadbands are about 24-32% larger than the values reported in Table 6 of Li and Barker  (2005), this also being inconsistent in implication with Fig. 5. However, these estimates do not account for the non-linear impact of the strong spectral variation of absorption



cross-sections from the GEISA database at lower wavelengths in the UV spectral range shown in Fig. 7. Effective band cross-sections from the GEISA spectrum were also calculated for each spectral region for irradiances at the surface using

$$c_{eff} = \frac{1}{N} \cdot \frac{\sum F_\lambda \Delta\lambda}{\sum F_\lambda e^{-Nc_\lambda} \Delta\lambda} \tag{4}$$

where $F_\lambda$ denotes the solar spectral irradiances in W/(m$^2 \cdot$ nm), $c_\lambda$ are the absorption coefficients set for a reference temperature

and pressure of 223 K and 100 mbar, and the $N$ is a total column ozone of $8.07 \times 10^{18}$ molecules/cm$^2$, equivalent to 300 DU. The numerator is equivalent to deriving broadband average solar fluxes from equally weighting values over all wavelengths as in the previous paragraph. The effective cross-section estimates calculated from Cloud-J for the two lowest UV bands (280-294 nm and 294-310 nm), with values of $1.09 \times 10^{-18}$ and $2.24 \times 10^{-19}$ cm$^2$/molecules respectively, are now instead smaller by 31% and 18% relative to the cross-sections referred in Li and Barker (2005) implying larger Cloud-J irradiances; values are larger

for the higher wavelength bands by 18% and 9%. The impact of these differences is made stronger for the lower bands as their absorption cross-sections are larger than for the higher bands by an order of magnitude or more; absorption by ozone in the higher bands is comparatively much weaker. The implied tendency is now in agreement with Fig. 5. This suggests weaker atmospheric attenuation at least from using the GEISA cross-section dataset instead of the broadband absorption cross-sections associated to the correlated-$k$ approach. Taking the spectrally dependent cross-sections and solar fluxes used with Cloud-J as

more reliable references, then one or both elements of the broadband cross-section and solar flux pairs associated to Li and Barker (2005) and GEM for the lower bands could be considered less optimal for determining irradiances at the surface. This stance is supported by the better agreement, for non-polar stations in Sect. 2.2, between the Cloud-J and Brewer sample spectra especially for the dominant 295-310 nm band.

Considering the above analysis of the differences in broadband irradiances shown in Fig. 5, scaling of the GEM irradiances to

the Cloud-J broadband irradiances was applied as functions of the irradiance values for each spectral band. While contributions to the UV index from the 280-294 nm band itself could be neglected for total column ozone above roughly 150 DU, scaling functions for this band were still generated since the band value is used in the spectral interpolation to higher wavenumbers for the second UV index model of Sect. 3.1. Also, scaling for the two highest UV bands is not essential and was done here for completeness. Fits were generated using the 7-day contributions for the total, direct, and diffuse irradiances of the four

bands under clear-sky conditions (23-29 August 2015). The scaling functions are provided in Table 1. The correlation of the broadband Cloud-J and the scaled GEM total irradiances obtained for clear-sky conditions are provided in Fig. 8.

### 3.1.2 UV index from broadband irradiances

The UV index fitting based on the Sect. 2.3 integral approach applied to GEM scaled broadband irradiances provided reference positions of 285.3, 302.7, 320.3, and 379.4 nm for bands 1 through 4, respectively, while the straight forward linear fit yielded:

$$UVI = 11.03 I_{294\text{-}310} + 0.084 I_{310\text{-}330} + 0.029 I_{330\text{-}400} \tag{5}$$

where the first coefficient was derived analytically as mentioned in Sect. 2.3. Most of the sensitivity to ozone variability is typically reflected in $I_{294-310}$ as absorption from ozone is comparatively weaker for the upper wavelength bands. Reductions



in column ozone by 20% from 300 DU imply changes of about 38%, 8.6%, and 0.15% in UV index from the last three terms, respectively, when the sun is directly overhead.

Differences of the clear-sky UV index field between the Cloud-J and resulting GEM values are shown in Fig. 9 and are found to be typically less than 0.2 for both the integration (upper panel, a) and linear fit (lower panel, b) approaches. Over

North America, the resultant UV index values are usually smaller than the Cloud-J based values by 0.1 to 0.3. Both plots also demonstrate an extended circular region at high zenith angles in the Southern Hemisphere with positive differences reaching up to 0.5 in the South Pacific area. These larger differences are coincident with UV index values near the threshold value of 3 used in the least-squares minimization of the scaled GEM broadbands to the high resolution UV index field produced by Cloud-J. In addition, there are a sparse number of hot spots which are primarily confined to the Arctic and the high altitude regions of the

Western Cordilleras of North and South America. Here, the differences in the UV index range between 0.2 to an extreme of 2.4, where the largest differences are confined to a few isolated mountain peaks in Ecuador and the Southern Patagonian Ice Fields bordering Argentina and Chile. The source of the hot spots were determined to be originating from the diffuse component of the calculated surface irradiances, where it was ascertained that the cause was ultimately due to differences in the albedo values used by the GEM and Cloud-J models, where the GEM albedo values underestimate the snow/ice reflectivities in these

regions. UV surface reflectivities for snow/ice are typically >85% (Chadyšien and Girgždys , 2008), and are readily observed in the OMI monthly average surface reflectivities used by Cloud-J. Although the GEM albedo values for these same regions are also elevated, with respect to the surrounding terrain, they are typically smaller as compared to the OMI-based climatology by 35-50%.

Curiously, there exists a notable cold spot in the plots of Fig. 9, and it too occurs in South America along a large barren

desert tract of the Andes mountains in northwestern Argentina, northern Chile, and southwestern Bolivia. Here, the GEM model indicates that surface reflectivities are elevated to values ranging from 60-75%, much higher than those associated with the snow/ice albedos representing the Southern Patagonian Ice Fields. OMI, on the other hand, produces reflectivities of only 10-15%, making little distinction with the surrounding landscape. Further investigation reveals that this region is variably snow-covered during the winter months of the Southern Hemisphere, where the presence of snow is not consistent throughout

the month or from year-to-year. During the 23-29 August 2015 analysis period used our study, this corresponding region of the Andes was covered under a fresh layer of snow. This observation is corroborated by both snow depth (SD) data obtained from the GEM model, and through visual confirmation using imagery data provided by the Moderate-Resolution Imaging Spectroradiometer (MODIS) instruments onboard the Aqua and Terra satellites (https://worldview.earthdata.nasa.gov/). Since the OMI albedo data represents monthly mean reflectivities over a 5 year period (2005-2009), it is unsurprising that a variable

presence of snow in this region creates disparities with the long-term averaged values recorded by OMI. The averaging would result in an underestimation in the OMI reflectivites, thus creating the observed cold spot seen in Fig. 9(a and b).

Figure 10 shows the resultant direct correlations between clear-sky UV index values obtained from the high resolution effective spectra versus those from the broadband Cloud-J and GEM irradiances for both the integration approach (a) and direct linear fit (b) for the data corresponding to the 7-day contributions over North America and the Arctic on August 23-29,

2015, at 18 UTC. The integration approach, used to weight the scaled GEM broadbands (cyan), show an excellent agreement





with the UV Index calculated using the Cloud-J broadbands (purple) where the slope of the curves, m, and associated Pearson correlation coefficients, R, are at unity. The resultant differences in UV indices from the high spectral resolution irradiances and the resultant GEM broadband irradiances are typically within 0.2 with a root mean square relative error in the scatter of ∼5.3% for clear-sky conditions. The UV indices calculated using the direct linear combination fitting of the GEM broadbands

produce similar results with a root mean square relative error in the scatter of ∼6.6% for UV index values larger 3.

### 3.1.3 Comparison to ground-based UV index measurements

Section 2.2 provided a comparison of simulated Cloud-J and measured Brewer sample irradiance spectra. The comparison with Brewer measurements is extended here to the clear-sky UV index and column ozone values from the GEM model short-term forecasts covering July and August of 2015. Figure 11 shows average differences in total column ozone between model short-

10 term forecasts and Brewer measurements in the range of 3.5 to 3.9% for the four non-Arctic stations with a decrease toward zero at higher latitudes for the two Arctic stations, Eureka and Resolute. This is consistent with column ozone differences stated in Sect. 2.2.

The UV index average values corresponding to the column ozone values were generated from the model short-term forecasts using the simplified spectral integration approach. The average UV index differences between the model forecasts and the

15 Brewers are -2.5 to -5% for the non-Arctic stations, partly explained by the differences in column ozone, to 6-9% for the two Arctic stations. The larger percentages for the two Arctic stations partly reflect relative increase in contribution from irradiances for bands above versus below 310 nm at higher solar zenith angles combined with the larger irradiance differences with Brewers above 310 nm mentioned in Sect. 2.2. This difference is not inconsistent with the 7.5% overestimation of the Cloud-J 310-325 nm band irradiance indicated in that same section and for which no scaling currently has been applied. The

20 source of differences of the UV index comparisons between the non-Arctic and Arctic stations in the overlap 50-60 degree region of Fig. 11 is not known. Still, considering the small UV index values at high solar zenith angle larger than 50 degrees, these translate to absolute differences with Brewers of less than 0.4.

The negative differences in UV index of -2.5 to -5% for the non-Arctic stations differ from the overall slightly positive differences of Cloud-J irradiances from Fig. 4 based on the five cases at 18 UTC for each station. Potential contributing sources

of differences are the residual errors from the fits for irradiances and for the UV index, the latter having been performed considering only values larger than 3; Fig. 9 indicates roughly -0.1 to -0.3 differences between GEM and Cloud-J over much of Canada. Reducing model ozone biases would improve the agreement with clear-sky Brewer UV index values by a few percent for UV index values above 3-4 and or solar zenith angles below 50-60 degrees. Additionally incorporating the 0.93 scaling factor correction alluded to in Sect. 2.2 would bring these differences back to about -5 to -7%, or roughly -0.3 to -0.5, while

improving results for the Arctic Stations.

### 3.2 Cloudy-sky conditions

As described in Sect. 1.2, the Cloud-J model possesses a number of options for the treatment of clouds in its radiative transfer calculations. Cloud-J broadbands were produced for each of the cloud options representing cloudy-sky conditions, 2-8, using





the GEM parameters for liquid and ice water partial column amounts of each model layer in the presence of clouds and the associated cloud fractions, which are required input for Cloud-J. The simulated broadbands produced by Cloud-J for each cloud option were then compared to the GEM analogs to determine which Cloud-J cloud flag produces output that best reproduces the GEM cloud-sky surface irradiances.

Prior to performing the comparative study, it was recognized that fundamental differences existed between Cloud-J and GEM with respect to the handling of clouds, particularly with respect to the scattering of light with parameters specific to water droplet/ice crystal size. Unlike GEM, the Cloud-J model does not specifically differentiate water droplets and ice crystals into different size bins and determine the scattering contribution accordingly. Instead, for water, an average droplet size is determined for the total water content in a particular model layer depending on the temperature and pressure associated with

the model layer. Ice crystals are not differentiated by size at all, only by crystal shape (hexagonal, amorphous), which is also determined by the given temperature and pressure of the model layer. Ultimately, it was determined that Cloud-J cloud option 3 produced cloudy-sky surface irradiances that best emulated the GEM analog. This option was therefore applied for the UV index comparisons in this section.

The estimation and evaluation of the UV index estimated under cloudy conditions in this study has been limited to the

consideration of two points. One is whether or not the UV index equations derived from clear-sky conditions are appropriate for cloudy conditions. The other is determining the level of impact of radiative transfer differences in the treatment of clouds on differences in derived UV index values.

The validity of the clear-sky UV index equations for cloudy conditions was tested using Cloud-J simulations. The clear-sky equations were applied to the Cloud-J broadband irradiances for comparison to the UV index values derived from the high

resolution Cloud-J spectra for the actual sky conditions from GEM-LINOZ, the latter being a mixture of clear-sky and cloudy-sky conditions. It was found that the equations derived for clear-sky conditions and applied to cloudy conditions with Cloud-J broadbands give essentially the same results as the UV index values from the high resolution spectra, i.e., no visible scatter about the diagonal is observed for the corresponding differences in Fig. 12a. Therefore, these equations would also be valid under cloudy conditions and do not require further adjustment.

The remainder of this section examines the impact of differences in cloud radiative transfer. Figure 12a shows the analogous correlations of the UV Index fields generated from the Cloud-J and GEM broadbands under all sky conditions using the Cloud-J cloud option 3. The weighting was performed using the values obtained though the integration approach of the GEM broadbands under clear skies. Weighting of the all sky broadbands using the values obtained from the linear fitting approach produce similar results. The overall correlation of the Cloud-J data is in fairly good agreement with the GEM data, but there

is an overall increase in error between the two data sets with increasing values of the cloud fraction. To better visualize the distribution density of the correlation, a density plot is also provided in Fig. 12b. We observe that the vast majority of points fall along the regression line, and ultimately represent those surface irradiances under cloudless, clear-sky, conditions. Deviation from the regression line typically increases with increasing cloud cover. Overall, the resultant differences in UV indices from the high spectral resolution irradiances and the resultant GEM broadband irradiances are similar under cloudy conditions with

light to moderate clouds, having a relative error comparable to the clear-sky counterpart, but under strong attenuation due to



clouds, a substantial increase in the root mean square relative error of up to 33% is observed due to differing cloud radiative transfer models for UV index values of 1 or larger.

Figure 13 contains a series of probability density plots to visualize the dependence of differences in surface irradiances on cloud cover for the 330-400 nm band. Relative differences are observed between the Cloud-J and GEM surface irradiances under unattenuated, clear-sky conditions, as well as for different total effective cloud fraction intervals. To filter for cloud cover, the GEM variable for total effective cloud cover (ECC) was used. ECC is employed in the plots to display the relative differences of the GEM and the Cloud-J irradiance values for a given range of cloud cover from clear-sky (0.0) to completely overcast (1.0). Only surface irradiances pertaining to zenith angles < 70°were included to remove larger systematic relative differences at high zenith angles where irradiance values are smaller. The Cloud-J cloud option 3 is used to calculate cloud attenuation in all cases. Output from two different settings of the GEM radiative transfer package for cloudy skies are separately provided for Cloud-J simulations and compared to the corresponding GEM irradiances.

The modification made to the GEM code from its reference settings of Sect. 1.1 was to increase the overall size of the effective radii for the ice clouds from a constant of 15 $\mu$m to values in the range of 20-50 $\mu$m to determine if it made any difference in relation to the Cloud-J output. As noted earlier in this section, Cloud-J does not differentiate between particle sizes in ice clouds. In the plots, we observe the increase range of relative differences with increasing cloud cover where differences can reach as high as 100% and above where the cloud fraction is $\geq 0.7$ (Fig. 13d). This implies, that different cloud radiative transfer settings or models can result in very large differences in UV index in the presence of optically thick clouds. Also notable, is the overall improvement on the left-hand side of the distributions when the ice particle size was increased. This illustrates the sensitivity of irradiances to cloud related model parameters. To quantify this sensitivity, the percentage contribution of the total discrete densities are compared for the relative differences in the ranges of -0.2 to 0.2 for cases representing $0.3 \leq$ ECC $< 0.7$ (moderate to heavy cloud) and ECC $\geq 0.7$ (heavy cloud to completely overcast) conditions, (Fig. 13c) and (Fig. 13d), respectively. Under moderate cloud to heavy cloud cover, the density distributions are similar in nature, where the percent contributions for both the modified and unmodified versions of the GEM model are ∼77%. For heavy cloud to completely overcast skies, there is a marked difference in the percent contributions. The unmodified GEM model cloud scheme produces a distribution where 50% of the discrete density is located within the -0.2 to 0.2 range for the absolute relative differences. Using the modified scheme, this value is increased to 62% stemming from more relative differences of smaller absolute size. These results and percentages provide some general sense of the potential uncertainties of the UV index values given possible uncertainties in the accuracy of the cloud radiative transfer models.

## 4 Conclusions

A successful optimization of UV index determination from broadband irradiances was performed. The Cloud-J v7.4 radiative transfer model was adapted to provide high spectral resolution surface irradiances in the UV, 280-400 nm. The high resolution output from Cloud-J is used to evaluate ECCC's GEM forecast model broadband irradiances under clear-sky conditions and to optimize the determination of the UV index using these coarse resolution spectral broadbands.




The optimization is achieved by creating simulated broadbands using Cloud-J for direct comparison with the GEM broadband irradiances to generate sets of scaling functions to calibrate the GEM values to the Cloud-J output. The scaled GEM broadbands are weighted accordingly such that the global UV index field produced using the coarse resolution broadbands subsequently replicate the high resolution UV index field calculated from Cloud-J. Further optimization with the current setup

could still be performed, such as excluding outlier differences and focusing over land areas in the fits, and further exploring the differences with the Brewer UV irradiance spectra and UV index values. The comparison with Brewer data for clear-sky conditions suggests potentially remaining systematic UV index differences up to about 0.3 to 0.5 in magnitude when the surface reflectivities are sufficiently representative.

It was established that equations for the UV index calculation determined from clear-sky conditions are also applicable to
cloudy conditions. However, as is to be expected, the quality of the UV index values strongly depend on the accuracy of the representation of clouds and, as implied in the limited evaluation of Sect. 3.2, on the accuracy of the cloud radiative transfer model. With formulations as developed here, the improvement of the quality of the UV index would follow the improvement in accuracy of these factors.

Outlier differences in UV index values under clear-sky conditions exemplified the relevance of using sufficiently representa-
tive surface reflectivities for snow and ice covered surfaces. Other factors, such as changes in the applied aerosol climatology or adjustments in the clear-sky irradiance calculation model might potentially warrant a revisiting of the fit coefficients.

The model simulations from Cloud-J, GEM, and similarly from other numerical prediction models, pertain only to the consideration of atmospheric columns directly overhead. While the solar zenith angle is reflected in the overhead column attenuation, the atmospheric conditions along the slanted viewing column may differ thus affecting the actual surface irradiances
and UV index. Moreover, for non-uniform cloud opacity, cloud scattering from various directions is unlikely to be correctly reflected from the overhead column or the solar viewing column alone. Accounting for these aspects, which is beyond the scope of this study, could further improve the accuracy of UV index forecasts.

*Code and data availability.*   The availability of the Cloud-J v7.4 radiative transfer model, and the various data sets used in the model modifications to calculate high-resolution surface irradiances including the TOA solar spectrum, $O_3$ cross-sections, surface reflectivities, and
Rayleigh scattering parameters are detailed in Sect. 2 of this publication. The output for the GEM forecast data and GEM-LINOZ $O_3$ fields are saved with an in-house binary file format; this in-house, binary file format is used to store gridded data from numerical weather and chemical prediction models, objective analyses and geophysical fields. Code changes made to Cloud-J to make use of such files takes advantage of in-house libraries. Selected data from these files, which can be reproduced in other desired formats, and related diagnostic results can made available upon request.

*Competing interests.*   The authors declare that they have no conflict of interest.





*Acknowledgements.* The authors would like to thank the Natural Sciences and Engineering Research Council of Canada (NSERC) for supporting K. A. Tereszchuk through the Visiting Fellowships in Canadian Government Laboratories Program (Grant: 462244-2014)), Michael Prather of the University of California, Irvine, for information on usage of Cloud-J, Quintus Kleipool of the Royal Netherlands Meteorological Institute for providing the solar spectrum, Vitali Fioletov and Akira Ogyu from ECCC regarding information on Brewer measurements,

5   Jean de Grandpré and Irena Ivanova (ECCC) for assistance in use of the GEM-LINOZ model, and Louis Garand (ECCC) for suggesting use of the GEM broadband irradiances for UV index determination.



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





**Table 1.** Sets of scaling functions to calibrate the GEM UV broadbands to emulate the simulated broadbands produced by Cloud-J. Functions were obtained for total surface irradiances and also their direct and diffuse components.

| Wavelength range | GEM UV broadband scaling functions | | |
|---|---|---|---|
| | Total irradiance (W/m$^2$) | Direct component (W/m$^2$) | Diffuse component (W/m$^2$) |
| 280-294 nm | $f(x) = \begin{cases} 0.551x^{0.615} & x \le 7.5\text{x}10^{-6} \\ 0.723x^{0.637} & x > 7.5\text{x}10^{-6} \end{cases}$ | $f(x) = \begin{cases} 0.558x^{0.625} & x \le 4.0\text{x}10^{-6} \\ 1.052x^{0.677} & x > 4.0\text{x}10^{-6} \end{cases}$ | $f(x) = \begin{cases} 0.225x^{0.582} & x \le 3.0\text{x}10^{-6} \\ 0.239x^{0.586} & x > 3.0\text{x}10^{-6} \end{cases}$ |
| 294-310 nm | $f(x) = \begin{cases} 1.109x^{0.659} & x \le 0.2 \\ 1.268x^{0.748} & x > 0.2 \end{cases}$ | $f(x) = \begin{cases} 1.004x^{0.679} & x \le 0.1 \\ 1.207x^{0.760}+ & x > 0.1 \end{cases}$ | $f(x) = \begin{cases} 0.746x^{0.611} & x \le 0.1 \\ 0.975x + 0.087 & x > 0.1 \end{cases}$ |
| 310-330 nm | $f(x) = 0.973x$ | $f(x) = 1.048x$ | $f(x) = x^{0.892}$ |
| 330-400 nm | $f(x) = 0.993x$ | $f(x) = 1.031x$ | $f(x) = x^{0.970}$ |

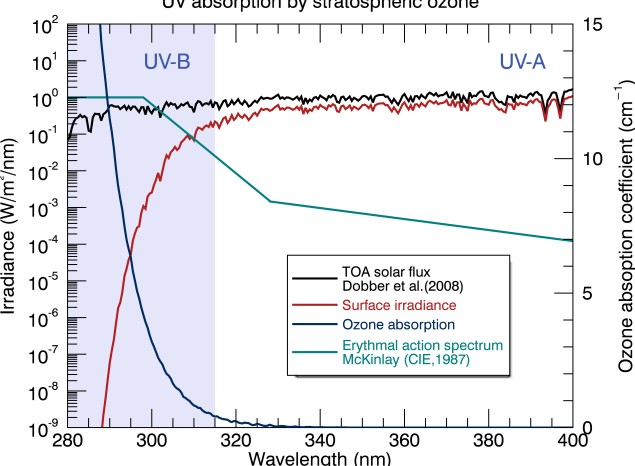

**Figure 1.** Sample UV irradiance spectrum at the Earth's surface on a clear summer day (averaged and sampled over 0.5 nm intervals). Stratospheric (O₃) is the primary species which serves to absorb UV radiation in the atmosphere (blue curve). The Huggins/Hartley band system of O₃ attenuates the radiative flux (black curve) by several orders of magnitude in the UV-B region. The product of the absorption cross-section and the top-of-atmosphere flux gives the resultant incoming irradiance at the surface (red curve). The erythemal action spectrum (green curve), demonstrates the increasing susceptibility of human skin to epidermal damage (erythema).



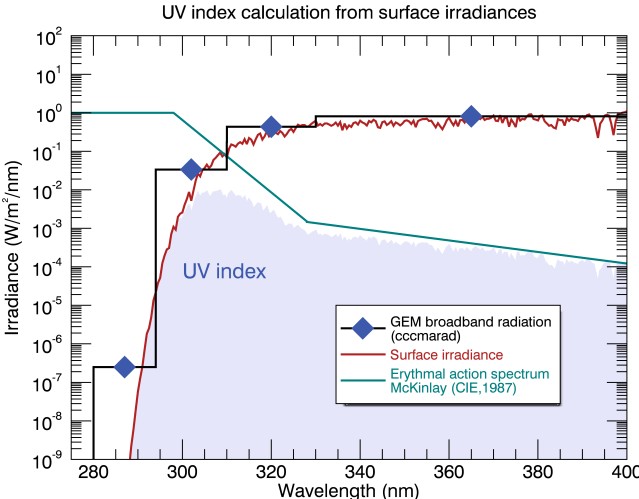

**Figure 2.** The UV index is defined as the integral of the erythemally weighted irradiance spectrum (shaded region), produced from the product of the surface irradiance (red curve; see Fig. 1) and the erythemal action spectrum (green curve), over the UV-A and UV-B spectral range. The result is then multiplied by a scaling factor $(25 \ mW/m^2)^{-1}$ to create a numerically convenient value for the index. Also depicted are the corresponding irradiances for the GEM broadbands divided by their respective bandwidths.

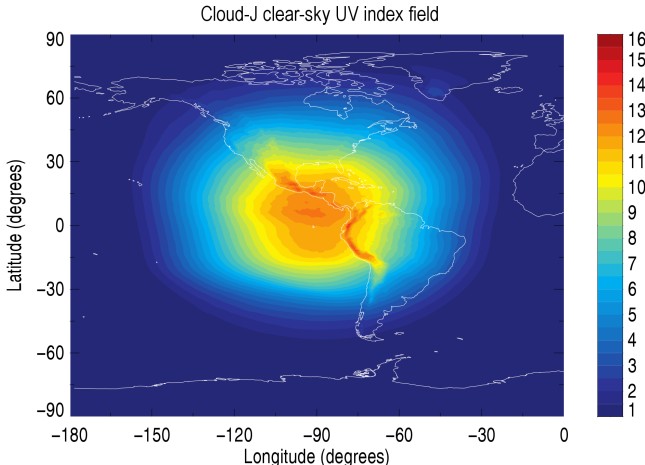

**Figure 3.** Cloud-J clear-sky UV index field produced using GEM 6h forecast data with the OMI and GEISA spectral parameters detailed in Sect. 2.1. The UV index field was generated from a 7-day average of spectral irradiances produced from 23-29 August 2015 at 18h UTC.







**Figure 4.** Cloud-J clear-sky surface irradiances compared to *in-situ* Brewer measurements obtained from six measurement stations belonging to ECCC's ozone monitoring network. Plotted are 5-day averages for 18h00 UTC of Brewer spectral irradiances (red curve) and the associated Cloud-J irradiances (light blue). The Cloud-J irradiances shown here were calculated with the Dobber et al. (2008) TOA spectrum averaged over 0.5 nm intervals with a sampling resolution also of 0.5 nm.



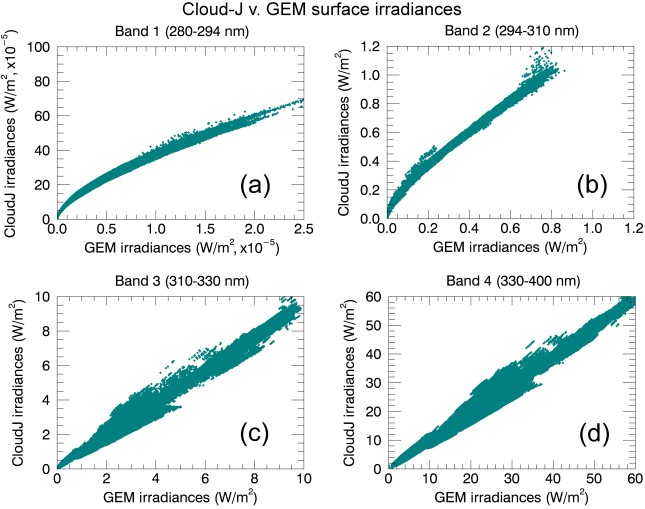

**Figure 5.** Correlation of GEM and Cloud-J total surfaces irradiances for clear-sky conditions. The GEM UV broadbands are compared to simulated broadbands produced by integrating the high resolution Cloud-J output over the same spectral regions. Presented are the individual, 7-day irradiance contributions from 23-29 August 2015.

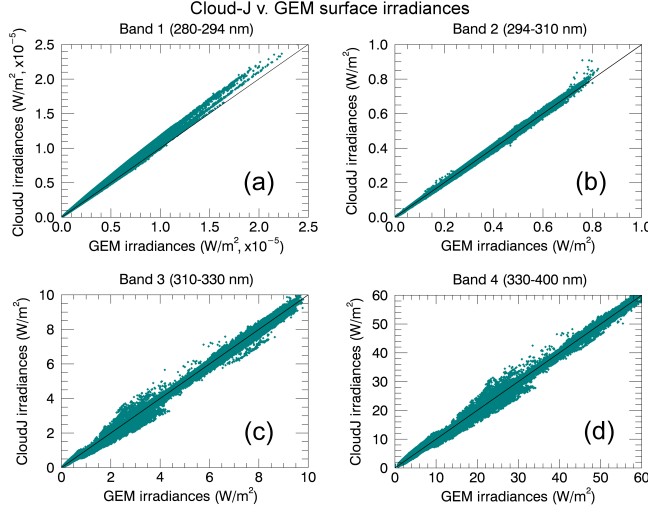

**Figure 6.** GEM broadband surface irradiances compared to simulated irradiances generated with Cloud-J, where the Cloud-J calculations were performed using the broadband absorption cross-section and TOA solar fluxes associated to the correlated-*k* scheme used by GEM for each UV sub-band. Correlations represent the single day irradiance contribution for 23 August 2015.

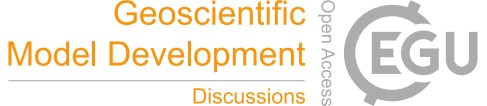



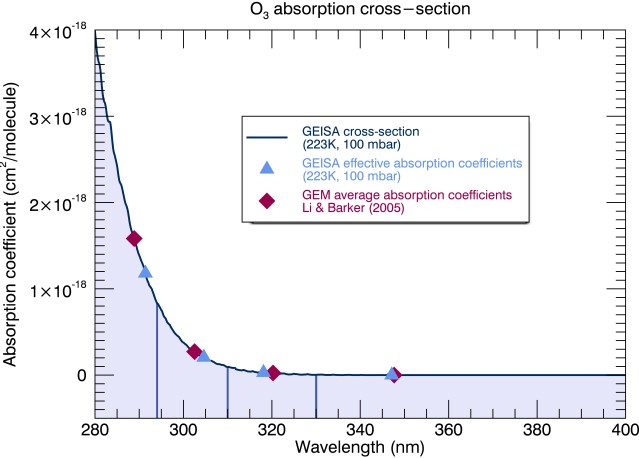

**Figure 7.** GEISA ozone absorption cross-sections measured at a temperature and pressure of 223 K and 100 mbar, respectively. Overlaid are the effective absorption coefficients calculated from the GEISA cross-section, as described in Sect. 3.1.1, and the GEM average absorption coefficients for each representative UV broadband.

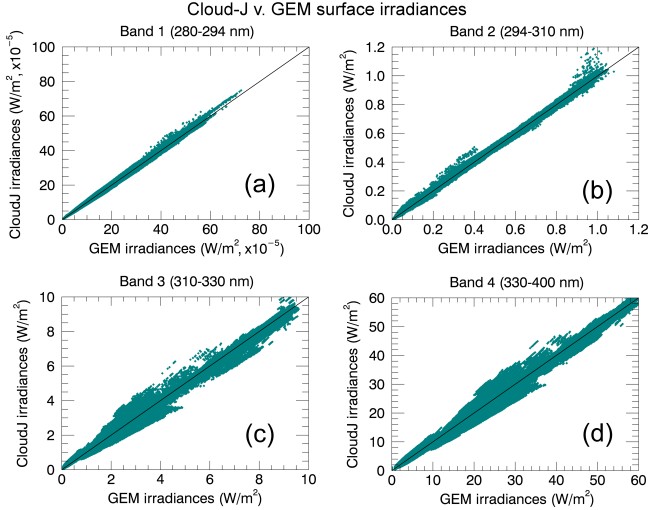

**Figure 8.** Calibrated GEM broadbands, corrected using the total irradiance scaling functions found in Table 1, compared to the simulated GEM broadbands produced by Cloud-J.





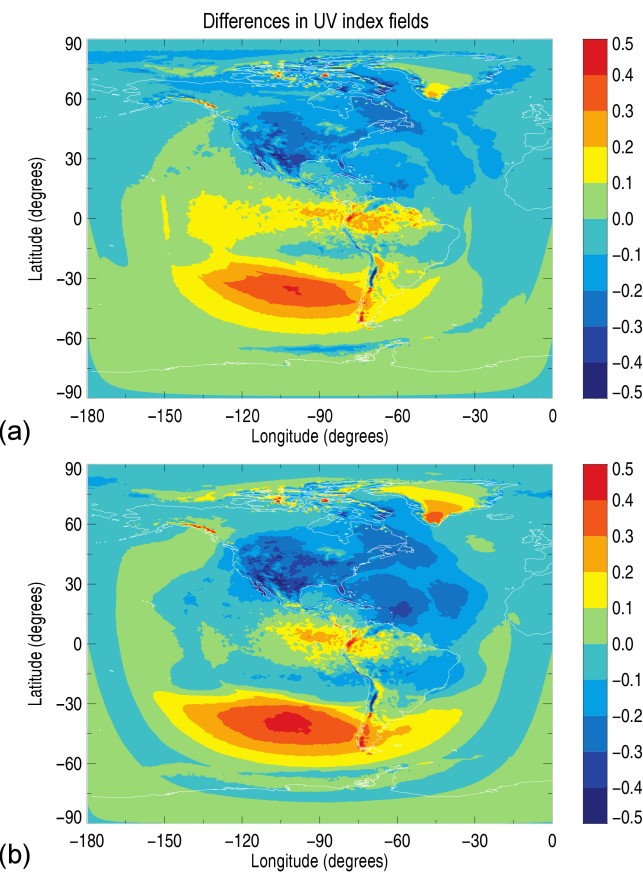

**Figure 9.** Differences in the UV index field produced from the scaled and weighted GEM irradiances compared to the field produced using the high resolution Cloud-J irradiances for the integration approach and linear fit, representing plots (a) and (b) respectively.




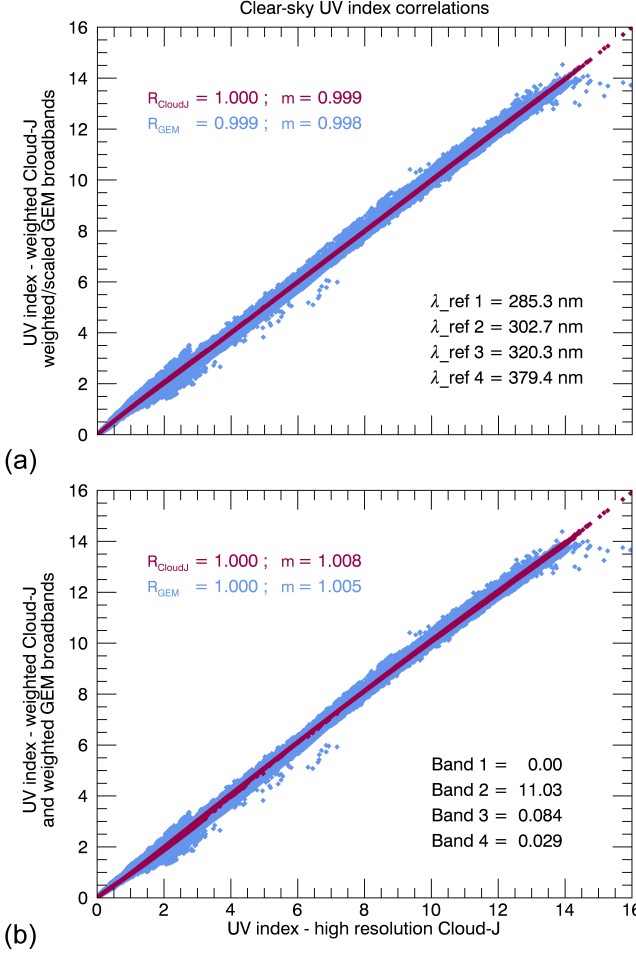

(a)

(b)

**Figure 10.** Correlation of the UV Index fields generated from the Cloud-J (purple) and GEM (blue) broadbands. Results from the least-squares minimization using the integration approach (upper panel, a) produced reference positions of 285.3, 302.7, 320.3 and 379.4, respectively for each UV sub-bands. Minimization performed using the direct linear fitting method (lower panel,b) produced coefficients of 11.03, 0.084, and 0.029 for bands 2 through 4, respectively, where the weighting for band 1 was intentionally fixed to a value of zero.





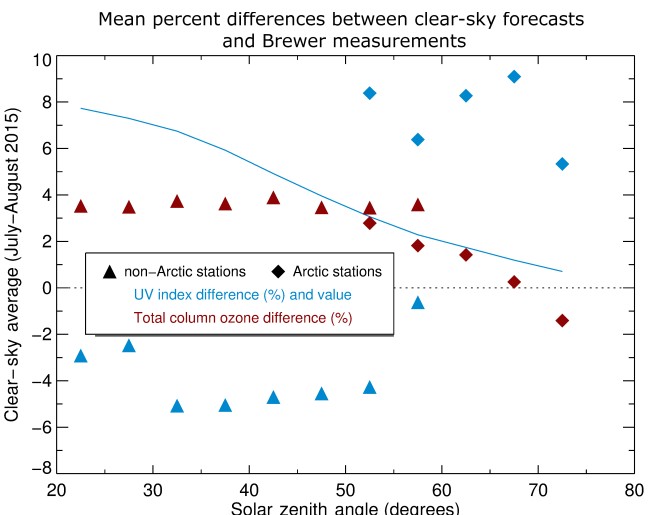

**Figure 11.** Average UV index and total column ozone relative differences between the model forecasts and Brewer measurements as a function of solar zenith angle for daytime clear-sky to lightly cloudy conditions for both sets over July and August 2015. This is accompanied by the corresponding average UV index values. The averages are over 5 degree intervals in solar zenith angles over the two Arctic stations (Eureka and Resolute) and four non-Arctic stations (Churchill, Edmonton, Saturna and Toronto) of Fig. 4. The resultant numbers of averaging points per bin range from 30 to 1002 with statistical outliers having been removed in final averages. Model short-term forecasts with output for station locations every 12 minutes were generated from weather and ozone analyses at 00, 06, 12, and 18 UTC.

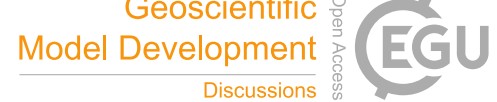



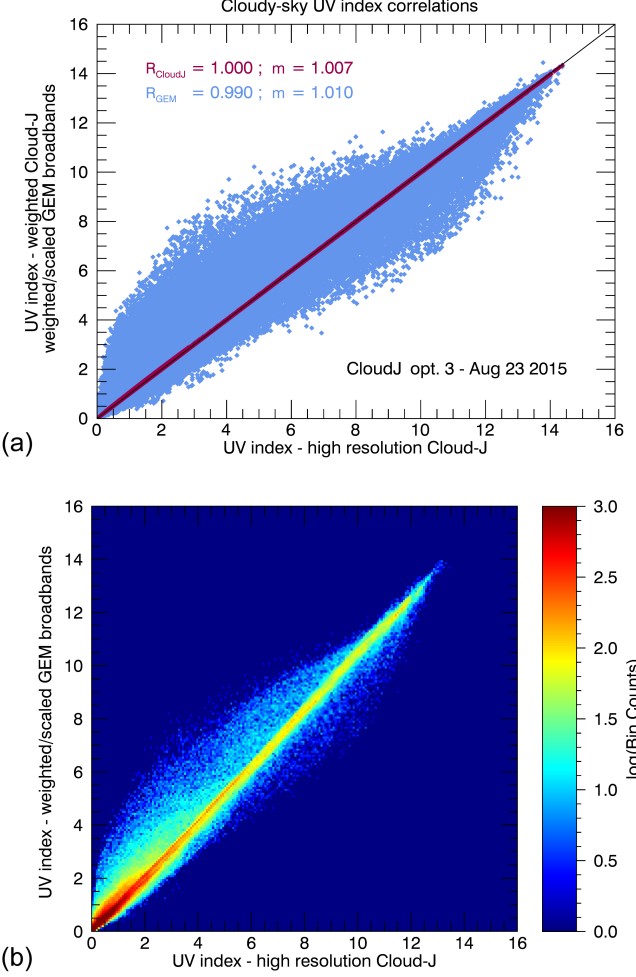

(a)

(b)

**Figure 12.** Analogous correlations of the UV Index fields generated from the Cloud-J (purple) and GEM (blue) broadbands under cloudy-sky conditions using Cloud-J cloud flag Option 3 in the comparison. The upper panel, a, presents the direct linear correlations of the UV index calculated using the GEM and Cloud-J broadbands relative to the high resolution output produced by Cloud-J using the same scaling functions and weighting determined through the integration approach under clear-sky conditions. The lower panel, b, is a density plot of the correlation of the UV index calculated using the GEM broadbands compared to the Cloud-J, high resolution, UV index field depicted in the upper panel.



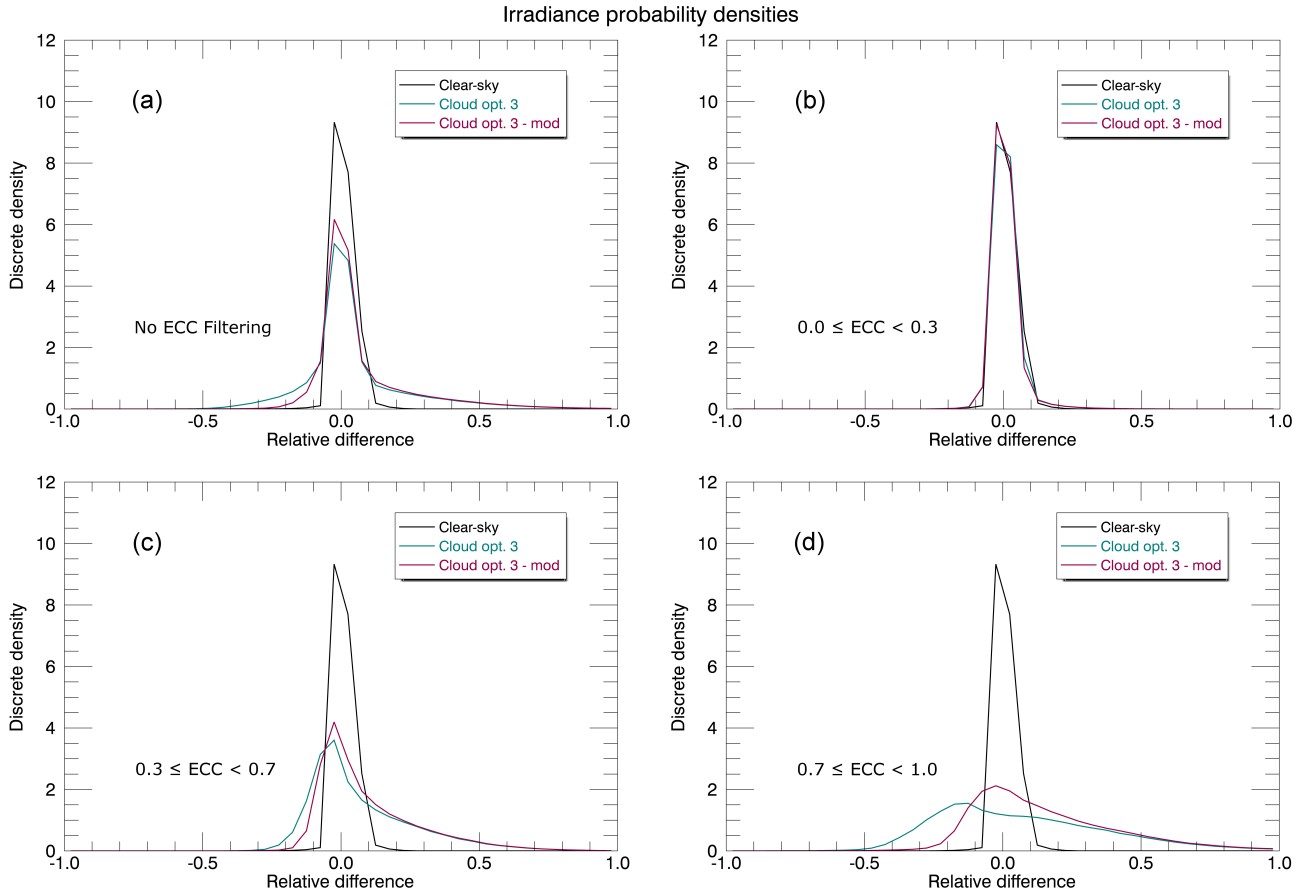

**Figure 13.** Irradiance probability density plots demonstrating the dependence of the 330-400 nm surface irradiances on cloud cover (ECC). Plotted are the relative differences between the Cloud-J and GEM surface irradiances under unattenuated, clear-sky conditions (black), cloudy-sky where the Cloud-J Option 3 cloud flag is used to calculate cloud attenuation (green), and a modified version of the GEM model output for cloudy skies compared to the Cloud-J data employing the Option 3 cloud flag (purple). The modification made to the GEM code was to change the effective radii for the ice clouds to determine if it made any difference relative to the Cloud-J output. In all four plots, a solar zenith angle filter was applied, where only surface irradiances pertaining to locations where zenith angles <70°are used. A secondary filter for varying total effective cloud cover is employed in the plots to display the relative difference in irradiances for a given range of cloud cover from clear-sky (0.0) to completely overcast (1.0).