# Peer review of "Optimizing UV Index determination from broadband irradiances"

_Geoscientific Model Development, 2017_

## Referee Comment (RC1) · Anonymous Referee #1 · 31 Dec 2017

This paper is a thorough study of utilizing the radiation output from a global weather forecast model to directly provide the UV index values. As the authors describe this is not an easy task. In order to even think about doing such a task is dependent upon how many bands the radiation spectrum is divided up into such that there are enough bands in the UV part of the spectrum to provide adequate results when weighted by the erythemal action spectrum. The authors go into great lengths and detail to ensure that their Cloud-J radiation scheme will produce accurate results. They compare test cases with UV spectral observations from the Canadian Brewer network. They determine that the Canadian Global Environmental Multi-scale model (GEM) has deficiencies in the incoming solar spectrum as well as values generated by its radiation code in the smaller UV wavelengths. Modifications to the downwelling shortwave radiation by clouds is

always a difficult task and is key to getting the global energy balance perfected. This paper provides the necessary homework prior to the next step of generating UV index forecasts for various hours of the day for multiple days. At that time forecast errors of ozone, clouds, snow will need to be evaluated.

The following are general comments and questions: - I am not accustomed to seeing correlations expressed as percent, rather as a unitless values ranging from -1.0 to 1.0. - This is a rhetorical question: Which is preferred : 'UV index' or 'UV Index'? Even the WMO web page has a mixture of both. But its acronym is 'UVI' implying that 'index' is capitalized.

Comments, questions: P2, L20: Should be 'oxygen ($O_2$)'.

P2, L27: Expand on what is meant by 'more sensitive population'.

P3, L4: 'erythemal action spectrum (EAS)'

P3, L15 The UVI does not have to be an integer.

P3, L17 The tropics have high UVI values not just because the SZA is small, but also because the total column ozone there is also low compared to higher latitudes.

P3, L24 I would also cite the two WMO reports addressing the UVI and the 'Global Solar UV Index' publication by the WMO.

P4, L 6 Is there a reference for the GEM?

P4, L11 Provide a reference here at the initial mention of the Cloud-J model.

P4, L20 Is there a reference for the cccmarad RTM?

P4, L34 Is the ozone (total and mixing ratios) really generated separately? The next paragraph discusses how LINOZ scheme is used 'within' the GEM to generate ozone forecasts. P10, L7 Can you determine the elevation adjustment per kilometer to determine if the difference between the gridpoint elevation and the actual elevation is the

reason?

P10, L7 While doing the above can you determine if the elevation adjustment is equal at all UV wavelengths or wavelength specific?

P10, L12 Is there a plan to bias adjust the GOME observations to bring them more in line with the Brewer observations?

P12, L4 Typing error : "Simpson's rule"

P12, L15 Do you plan to 'correct' the GEM equivalent broadband absorption cross-section and the TOA solar fluxes to agree with the Cloud-J?

P13, L20 All of these adjustments or scalings will need to be revisited every time the GEM's radiation code is modified. Communication and collaboration between the authors and the GEM modelers needs to be strong such that these differences can be addressed and best corrected in the GEM so that the number of adjustments in the UVI computations is limited or eliminated.

P13, L31 The I294-310 is where the bulk of the erythemal weighted values come from. I would think that the coefficient (11.03) would be total column ozone and solar zenith angle dependent.

P14, L18 I presume that the actual total column ozone was used during this comparison and not the OMI climatology, then why wasn't the GEM's albedo used instead of the OMI albedo climatology? Does the GEM's albedo need to be corrected to the OMI's for 100% snow cover? Using the GEM's albedo would then eliminate the 'cold spot' discussed in the following paragraph. The purpose of these two difference plots should be to show the differences between the integrated and linear solutions, not the differences between each and the GEM.

P15, L8 What is meant by 'short term forecasts'? 6, 12, 24, 48 hour?

P15, L30 Instead of just using the 18 UTC observations and model output, other times

of the day could have been used to generate additional UVI and solar zenith angle determinations. Additionally, the range of total ozone values over Canada during July and August are rather small. Comparisons between model and observations could have also been done for April or May when the sun is relatively not too low in the sky but range of total column ozone values is much greater.

P16, L31 As % cloud amount increases so does the variability of transmission through the clouds. Such a plot in place of the density plot may better show the differences between the Cloud-J and the GEM all sky values.

It would be interesting to note if, via the Cloud-J model, there is a spectral dependence of UV adjustments upon cloud amount, type, altitude.

The point of the text discussion is that the GEM and the Cloud-J produce reasonably similar results under all-sky conditions. Is it known whether either is correct against real world observations from the Brewers or solar radiometers?

P18, L10 I gather this answers my previous question about spectral impacts upon cloud amount. Or else the impacts are accumulated in the band coefficients.

P18, L22 There are only so many aspects of solar radiation that can be accounted for. Hopefully, these additional aspects are second order and fall within the error bars of the UVI values.

Figure 11 The symbols and line need to be identified in the figure caption. The Y axis caption also needs to have '% difference' in it.

Figure 13 Add 'effective' to cloud cover (ECC) in first line of figure caption.

---

## Referee Comment (RC2) · Anonymous Referee #2 · 27 Jan 2018

General comments:

The manuscript presents an interesting proposal on how to optimize calculations of UV Index forecasts by making use of the broadband irradiances produced by a specific numerical prediction model. UV Index is an essential tool in the promotion of sun safety and the attempts to minimize the adverse health effects of UV radiation on people. Production of UV Index forecasts on a global scale and yet in a useful spatial and temporal resolution is a computationally intensive task. Attempts to optimize the calculation procedures are therefore highly welcome.

The manuscript also provides a contribution to the current understanding on the differences between the surface irradiances obtained by radiative transfer model simulations vs. those measured in situ by Brewer spectrophotometers.

[Figure]

In general, the paper is well-organized and reads reasonably well.

Specific comments:

Page 1 Row 1: In its current form, the abstract is quite long. The reader would appreciate a more concise abstract where the main objectives and major findings are summarized.

Page 3 Row 3: As regards the action spectrum for erythema, which is the basis for the UV Index, you refer to McKinlay&Diffey (1987) and CIE Technical Report (2014). However, Eq. (1) does not exactly comply with either of these. In the formulation given by McKinlay&Diffey (1987), there are no "smaller than" ("<") signs, only "smaller than or equal" ("<=") signs. This would cause a small jump at 328 nm - which you do not have in your curve in Fig. 1, so probably you are not using the action spectrum of McKinlay&Diffey (1987).

CIE Technical Report (2014) refers to ISO/CIE1999 and gives a piecewise function where the signs are like in your Eq. (1). However, the equation for the range 328 < lambda < 400 includes a term (140 – lambda), not (139 – lambda) in the exponent, as does your Eq. (1).

Please check which erythemally weighted action spectrum you are using and give a reference for that. An excellent description on the differences between the different erythemally weighted action spectra may be found, for instance, in Webb et al. (2011).

Reference: Webb, A.R., Slaper, H., Koepke, P. & Schmalwieser, A.W. 2011. Know your standard: clarifying the CIE erythema action spectrum. Photochemistry and Photobiology 87: 483-486.

Page 4 Line 1: You refer to Long (2003) in the context of UV Index forecasting practices worldwide. More recently, Schmalwieser et al. (2017) has also reported on UV Index monitoring practices in Europe. That work could be also worth referring to.

Reference: Schmalwieser, A.W., Grobner, J., Blumthaler, M., Klotz, B., De Backer,

H., Bolsee, D., Werner, R., Tomsic, D., Metelka, L., Eriksen, P., Jepsen, N., Aun, M., Heikkila, A., Duprat, T., Sandmann, H., Weiss, T., Bais, A., Toth, Z., Siani, A., Vaccaro, L., Diemoz, H., Grifoni, D., Zipoli, G., Lorenzetto, G., Petkov, B.H., di Sarra, A.G., Massen, F., Yousif, C., Aculinin, A.A., den Outer, P., Svendby, T., Dahlback, A., Johnsen, B., Biszczuk-Jakubowska, J., Krzyscin, J., Henriques, D., Chubarova, N., Kolarz, P., Mijatovic, Z., Groselj, D., Pribullova, A., Gonzales, J.R.M., Bilbao, J., Guerrero, J.M.V., Serrano, A., Andersson, S., Vuilleumier, L., Webb, A. & O'Hagan, J. 2017. UV Index monitoring in Europe. Photochemical & Photobiological Sciences 16: 1349-1370. Page 4 Line 26: "the total (clear+cludy) sky analog". It is not very clear to this reader what this means. Could you please rephrase?

Page 9 Line 6: You have chosen to use weekly (7-day) averages. Could you please explain to the reader why you have chosen averages calculated for a period of 7 days? Why not 5 days – or 10 days?

Page 9 Line 24: You examine 5-day averages of Brewer measurements. Could you please justify the use of 5-day averages? Why not 7-day averages here?

Page 9 Line 23. You remind the reader that a boxcar averaging window was used for the OMI composite TOA spectrum and point out that the slit function of a Brewer spectrophotometer is trapezoid-shaped. The Brewer spectra can be purged from the effects of the slit function by performing a deconvolution. Could you please briefly discuss on how much the different schemes, averaging with a boxcar window vs. convolution with a triangular slit function, may be estimated to affect to the spectra.

Page 16 Line 2: "The simulated broadbands". I think it should be "The simulated broadband irradiances". There are some other instances in the body text with the same kind of formulation where the actual physical quantity is missing, like on Page 16 Line 28: "all sky broadbands" or Page 14 Line 8: "GEM broadbands". Please add the name of the physical quantity wherever it is currently missing.

Page 17 Line 33: What is a "spectral broadband"? Please explain the term.

Page 18 (Conclusions). The reader would be extremely interested in any estimate on how much your approach would save computer time as compared to the current operational UV index forecasting. Would you please be able to give an estimate on that?

---

## Author Comment (AC1) · 27 Feb 2018

**Responses to Anonymous Referee #1**

Interactive comments on:

**Optimizing UV index determination from broadband irradiances**
**by**
**Keith A. Tereszchuk et al.**

Provided by: Anonymous Referee #1

Preamble: In addition to the changes made to the manuscript following the suggestions of the two reviewers, we also identified three minor issues that required performing a re-analysis of the data and also modifying comments regarding the comparison to Brewer measurements of Section 2.2. These changes did result in a small correction in the scaling and fit factors, but results remain essentially the same (other than improving the agreement with Brewer UV irradiances). These points are as follows:

1. An adjustment of the broadband wavelength boundaries from 280, 294, 310, and 400 nm to 280.11, 294.12, 310.70, and 400.00 nm (this was an error)

2. A correction in applying a moving boxcar averaging window covering $\pm 0.25$ nm about sampling points at intervals of 0.5 nm

3. A correction in the calculation of differences with the Brewer UV spectra in Section 2.2, resulting, most significantly, in a reduction of the mean percent differences for the 311-330 nm band from 7.5% to 2.9% (and implying related text changes).

The correction of few other typographical errors were also made.
The following are general comments and questions:

I am not accustomed to seeing correlations expressed as percent, rather as a unitless values ranging from -1.0 to 1.0.

The root mean square relative error calculation provides a more tangible, quantitative, overall representation of the errors and their scatter. Percentages were used to determine the absolute range of values that were encountered. While the Pearson correlation coefficient, R (ranging from -1.0 to 1.0), which provides a value which represents the "goodness" of the fit in a linear correlation, could also have been calculated and shown, the root mean square errors (and mean differences) were considered sufficient and more appropriate for our purpose, in addition to presenting the scatter plot themselves.

This is a rhetorical question: Which is preferred : 'UV index' or 'UV Index'? Even the WMO web page has a mixture of both. But its acronym is 'UVI' implying that 'index' is capitalized.

While, we are not aware of a definitive preference, one could choose to follow the convention of a capitalized 'index' considering the often used acronym 'UVI'. The format for using the lower case i in this work was simply done to avoid any possible conflicts with the manuscript composition guidelines with regard to figures, section titles, etc that have been detailed by the publisher, while trying to maintain consistancy throughout the paper. Having said that, we see now that there are 2-3 times where consistancy was not maintained. The manuscript has therefore been edited so that 'UV Index' appears using only the uppercase I and the term is treated as a proper name where the word index should be capitalized.

Comments, questions:
P2, L20: Should be 'oxygen (O2)'.

Correction made.

P2, L27: Expand on what is meant by 'more sensitive population.

Changed to more photosensitive populations. This refers to people who have extremely little melanin/skin pigmentation which provides a natural UV barrier. There are also a number of ailments where minimal doses of UV radiation can cause allergic reactions and severe burns. Increased photosensitivity can also present itself as a side effect for a variety of medications.

P3, L4: 'erythemal action spectrum (EAS)'

Correction made.

P3, L15 The UVI does not have to be an integer.

The word 'integer' has been removed.

P3, L17 The tropics have high UVI values not just because the SZA is small, but also because the total column ozone there is also low compared to higher latitudes.

Commentary appended to the manuscript as '... solar zenith angle and the total

column ozone are small.'

P3, L24 I would also cite the two WMO reports addressing the UVI and the 'Global Solar UV Index' publication by the WMO.

Citations added in a new paragraph immediately following equation 3. The bulk of that paragraph was previously in the abstract and moved following a request by the second reviewer to shorten the abstract.

P4, L 6 Is there a reference for the GEM?

Citation of 'Charron et al. (2012), and references therein,' added.

P4, L11 Provide a reference here at the initial mention of the Cloud-J model.

Citation of Prather (2015) added.

P4, L20 Is there a reference for the cccmarad RTM?

Citation of Scinocca et al. (2008) added.

P4, L34 Is the ozone (total and mixing ratios) really generated separately? The next paragraph discusses how LINOZ scheme is used 'within' the GEM to generate ozone forecasts.

Yes, the ozone analyses (ozone mixing ratio analysis fields) were generated separately from the weather analyses. This just means that a separate application of the variational assimilation with GOME-2 ozone data (described in the next paragraph of that section) was applied for ozone relative to the application of variational assimilation with weather data. This was done since the weather analyses had been generated previously. The ozone forecasts are generated by LINOZ within GEM. The process of assimilation improves on these forecast fields at regular time intervals of 6 hours, these improved fields are called analyses and are fed back to the model (to serve as new initial conditions) for it to generate the forecasts for the following time periods. The total column ozone is calculated (also in LINOZ) from vertically integrating the ozone mixing ratio profiles.

P10, L7 Can you determine the elevation adjustment per kilometer to determine if the difference between the grid point elevation and the actual elevation is the reason?

This could be performed but was not done. It is just one factor among many others that could have affected the comparisons and results over the different stations and there was no clear evidence to suggest that it was a comparatively important attribute to this effect. A small change in UV index in the order of roughly 1.5% only is estimated for a difference of 150 m. A related statement has been added to the text.

P10, L7 While doing the above can you determine if the elevation adjustment is equal at all UV wavelengths or wavelength specific?

An adjustment for elevation would require a wavelength specific correction. To demonstrate, the surface pressure was modified in Cloud-J for seven wavelengths in the 280-400 nm range in a given geographical location (Toronto, 29 Aug 2015) to

determine the percent difference in the surface irradiance contribution for each of the wavelengths at two different surface pressures. Pressures of 995 hPa and 1013 hPa were used, which correspond to an altitude difference of ∼150 m. The percent differences in irradiance are as follows: 280 nm (1.55%), 300 nm (1.05%), 320 nm (0.68%), 340 nm (0.54%), 360 nm (0.45%), 380 nm (0.38%), 400 nm (0.32%). In all, a disparity in altitude of up to 150 m would generate an error no greater than ∼1.5%.

P10, L12 Is there a plan to bias adjust the GOME observations to bring them more in line with the Brewer observations?

Yes. That work has been done (using another satellite data source that is in better agreement with Brewers) and is the topic of a separate journal paper to be submitted.

P12, L4 Typing error : "Simpson's rule"

Text Corrected.

P12, L15 Do you plan to 'correct' the GEM equivalent broadband absorption cross-section and the TOA solar fluxes to agree with the Cloud-J?

We only plan to scale the GEM broadband irradiances to emulate the general tendency of Cloud-J output as a function of the irradiance value as done in this paper and indicated at the end of Section 3.1.1. This takes the position that the OMI TOA solar spectrum in Cloud-J are deemed 'correct'. The sample effective broadband cross-section value was derived for irradiances at the surface and would not be valid for other levels (nor necessarily all differing ozone profiles). As well, choosing to

revisit or not the TOA solar fluxes (and or the applied cross-sections) would be the prerogative of those responsible for the model.

P13, L20 All of these adjustments or scalings will need to be revisited every time the GEM's radiation code is modified. Communication and collaboration between the authors and the GEM modelers needs to be strong such that these differences can be addressed and best corrected in the GEM so that the number of adjustments in the UVI computations is limited or eliminated.

Yes, should there ever be future pertinent modifications to the GEM radiation code, adjustments to the scalings and fits for UV Index determination would have to be made or considered depending on the significance of the change. We do not anticipate our work affecting the GEM radiation code development as such. A suite of programs has been created with a users' manual so that other collaborators/modelers can rerun the programs using updated GEM output files containing data using a newer/modified radiation scheme so new sets of scaling functions can be generated and the broadbands re-weighted accordingly.

P13, L31 The I294-310 is where the bulk of the erythemal weighted values come from. I would think that the coefficient (11.03) would be total column ozone and solar zenith angle dependent.

Considering Figure 9 and the related results, it was a pleasant surprise that constant scaling factors were sufficient for that equation to provide good UV Index values (e.g. errors/differences in UV Index of typically still within 0.2-0.3 with some exceptions), this in light of the erythemal weight varying significantly with wavelength for the two central bands. So the dependence of the UV Index on SZA and ozone with this equation is

sufficiently well captured through only the broadband irradiances themselves - as the calculation of broadband irradiances is dependent on SZA and the ozone profile (and, as such, total column ozone).

P14, L18 I presume that the actual total column ozone was used during this comparison and not the OMI climatology, then why wasn't the GEM's albedo used instead of the OMI albedo climatology? Does the GEM's albedo need to be corrected to the OMI's for 100% snow cover? Using the GEM's albedo would then eliminate the 'cold spot' discussed in the following paragraph. The purpose of these two difference plots should be to show the differences between the integrated and linear solutions, not the differences between each and the GEM.

The OMI albedo climatology was used because, with it, wavelength specific global fields for surface reflectivity could be used by Cloud-J to perform the high spectral resolution irradiance calculations. GEM uses a different approach for surface reflectivity where global albedo fields are represented by broadband (UV-NIR) effective values for reflectivity that have been differentiated within the model for specific surface types, of which contain the albedos for soil, glaciers, water, ice, and the aggregated value. For this work, it was therefore deemed more advantageous to use the OMI climatology for our purposes. The resulting larger differences such as in the 'cold spot' would have been removed/reduced using the same albedos. We were willing to accept retaining these differences considering the good agreement of the results elsewhere.

Whether the GEM albedos would need to be corrected to (or account for) the OMI-based values for 100% snow cover would then be the prerogative of the GEM model developers.

We think the purpose of the Fig. 9 needs to cover both aspects (differences with GEM-based values and differences between the methods). While this is implied at the beginning of the second paragraph of Section 3.1.2, a sentence has been added to mention that the integration approach fairs a bit better, i.e. 'The integration approach provides better agreement to Cloud-J, this by up to about 0.1-0.2 for some locations.'

P15, L8 What is meant by 'short term forecasts'? 6, 12, 24, 48 hour?

For this particular section, it refers to daytime 7.5 minute time steps (corrected from 12 minutes - a mixup between two forecast setups) covering up to 24 hours. The text at the beginning of this section has been changed to 'GEM model 24-hour forecast output at 7.5 minute intervals over successive twelve hour forecasts'. A similar change has been made in the last sentence of the Fig. 11 caption. In the section the data spanned solar zenith angles from morning to night. Other references to 'model short-term forecasts' in this section were changed to 'model output'.

P15, L30 Instead of just using the 18 UTC observations and model output, other times of the day could have been used to generate additional UVI and solar zenith angle determinations. Additionally, the range of total ozone values over Canada during July and August are rather small. Comparisons between model and observations could have also been done for April or May when the sun is relatively not too low in the sky but range of total column ozone values is much greater.

The 18 UTC field alone were used only in the other sections as clarified for the above point. While we did have forecasts relying on assimilated ozone data for the Summer already available, we did not have such forecasts for the Spring.

P16, L31 As % cloud amount increases so does the variability of transmission through the clouds. Such a plot in place of the density plot may better show the differences between the Cloud-J and the GEM all sky values. It would be interesting to note if, via the Cloud-J model, there is a spectral dependence of UV adjustments upon cloud amount, type, altitude. The point of the text discussion is that the GEM and the Cloud-J produce reasonably similar results under all-sky conditions. Is it known whether either is correct against real world observations from the Brewers or solar radiometers?

The purpose of the density plot was to make the point that there were many more occurrences of reasonably good agreement than large disagreements, this not being evident from panel of Figure 12. The previous P16 L32-33 lines were modified to '... along or near the regression line, largely, but not entirely, represent those surface irradiances under cloudless or light-cloud, conditions. The probability of deviation from the regression line typically increases with increasing cloud amount and opaqueness.' This is further illustrated by Figure 13 which shows the reduced agreement with increasing ECC (cloud fraction X (1- cloud transmittance)). The last sentence of that paragraph was not clear and out of place. It has been moved following the next paragraph introducing Figure 13 and clarified.

A plot of cloud amount itself versus variability (and or differences) of transmission through the clouds would provide a demonstration of the difficulty of correlating cloud amount alone to the impact of clouds on the UV Index - at least when cloud amounts are not that small, e.g. $\gtrsim$30-50%. While we prefer not embarking on this for this paper, it is worth considering in further examining/qualifying the impact of clouds (and model clouds) on the UV Index.

It should be noted that the Cloud-J calculations do take into consideration cloud scattering and absorption that is wavelength dependent and also accounts for water

droplet size and ice crystal type. Values are provided through the use of look-up tables.

We have not evaluated the accuracy of the model clouds in their resulting impact, characteristics, and coincidence of occurence relative to ground-based measurements (or even satellite base cloud measurements). We do not known the level of correctness of either cloud models. This is something of interest that is beyond the scope of this study.

P18, L10 I gather this answers my previous question about spectral impacts upon cloud amount. Or else the impacts are accumulated in the band coefficients.

The CloudJ calculations for cloud scattering and absorption are wavelength dependent using interpolated data obtained from look-up table parameters for water droplet size and ice crystal type.

P18, L22 There are only so many aspects of solar radiation that can be accounted for. Hopefully, these additional aspects are second order and fall within the error bars of the UVI values.

Hopefully, these geometry considerations do fall within the current uncertainty levels associated to the representation of spatially extended overhead clouds and their impact on the UV index. The treatment of water/ice clouds, particularly for non-uniform opacity and scattering within clouds, is one of the most challenging aspects to correctly manage within radiative transfer models.

Figure 11 The symbols and line need to be identified in the figure caption. The Y axis

caption also needs to have '% difference' in it.

Correction made.

Figure 13 Add 'effective' to cloud cover (ECC) in first line of figure caption.

Correction made.

---

## Author Comment (AC2) · 27 Feb 2018

**Responses to Anonymous Referee #2**

Interactive comments on:

**Optimizing UV index determination from broadband irradiances**
**by**
**Keith A. Tereszchuk et al.**

Provided by: Anonymous Referee #2

Preamble: In addition to the changes made to the manuscript following the suggestions of the two reviewers, we also identified three minor issues that required performing a re-analysis of the data and also modifying comments regarding the comparison to Brewer measurements of Section 2.2. These changes did result in a small correction in the scaling and fit factors, but results remain essentially the same (other than improving the agreement with Brewer UV irradiances). These points are as follows:

1. An adjustment of the broadband wavelength boundaries from 280, 294, 310, and 400 nm to 280.11, 294.12, 310.70, and 400.00 nm (this was an error)

2. A correction in applying a moving boxcar averaging window covering $\pm0.25$ nm about sampling points at intervals of 0.5 nm

3. A correction in the calculation of differences with the Brewer UV spectra in Section 2.2, resulting, most significantly, in a reduction of the mean percent differences for the 311-330 nm band from 7.5% to 2.9% (and implying related text changes).

The correction of few other typographical errors were also made.

Page 1 Row 1: In its current form, the abstract is quite long. The reader would appreciate a more concise abstract where the main objectives and major findings are summarized.

The abstract has been made more succinct.

Page 3 Row 3: As regards the action spectrum for erythema, which is the basis for the UV Index, you refer to McKinlay&Diffey (1987) and CIE Technical Report (2014). However, Eq. (1) does not exactly comply with either of these. In the formulation given by McKinlay&Diffey (1987), there are no "smaller than" ("$<$") signs, only "smaller than or equal" ("$\leq$") signs. This would cause a small jump at 328 nm - which you do not have in your curve in Fig. 1, so probably you are not using the action spectrum of McKinlay&Diffey (1987). CIE Technical Report (2014) refers to ISO/CIE1999 and gives a piecewise function where the signs are like in your Eq. (1). However, the equation for the range $328 <$ lambda $< 400$ includes a term $(140 - $ lambda$)$, not $(139 - $ lambda$)$ in the exponent, as does your Eq. (1). Please check which erythemally weighted action spectrum you are using and give a reference for that. An excellent description on the differences between the different erythemally weighted action spectra may be found, for instance, in Webb et al. (2011).

Reference: Webb, A.R., Slaper, H., Koepke, P. & Schmalwieser, A.W. 2011. Know your standard: clarifying the CIE erythema action spectrum. Photochemistry and Photobiology 87: 483-486.

The erythmal action spectrum that was originally intended to be used was the McKinlay&Diffey (1987) reference spectrum. This spectrum had been reported in a number of publications in the literature search of the UV Index as the benchmark erythemal

spectrum to be used in the calcualtion of the UV Index. Ultimately, the piece-wise function that was actually used was the one detailed in a NOAA reference article found here:

http://www.esrl.noaa.gov/gmd/grad/neubrew/docs/UVindex.pdf

The article cites their representation of the erythmal action function as being the one published by McKinlay&Diffey (1987). It appears that the NOAA article contains a typo in the wavelength limits that had not been noticed.

The UV calculations in this work and associated figures for the manuscript have been redone using the action spectrum detailled in the CIE Technical Report (2014). The jump referred to at 328 nm is present in the original Fig. 1 plot, but is not large enough to be discernible. The manuscript has been edited to explain the change in the function and reference has been made to the Webb et al. (2011) publication.

Page 4 Line 1: You refer to Long (2003) in the context of UV Index forecasting practices worldwide. More recently, Schmalwieser et al. (2017) has also reported on UV Index monitoring practices in Europe. That work could be also worth referring to.

Reference: Schmalwieser, A.W., Grobner, J., Blumthaler, M., Klotz, B., De Backer, H., Bolsee, D., Werner, R., Tomsic, D., Metelka, L., Eriksen, P., Jepsen, N., Aun, M., Heikkila, A., Duprat, T., Sandmann, H., Weiss, T., Bais, A., Toth, Z., Siani, A., Vaccaro, L., Diemoz, H., Grifoni, D., Zipoli, G., Lorenzetto, G., Petkov, B.H., di Sarra, A.G., Massen, F., Yousif, C., Aculinin, A.A., den Outer, P., Svendby, T., Dahlback, A., Johnsen, B., Biszczuk-Jakubowska, J., Krzyscin, J., Henriques, D., Chubarova, N., Kolarz, P., Mijatovic, Z., Groselj, D., Pribullova, A., Gonzales, J.R.M., Bilbao, J., Guerrero, J.M.V., Serrano, A., Andersson, S., Vuilleumier, L., Webb, A. & O'Hagan, J.

2017. UV Index monitoring in Europe. Photochemical & Photobiological Sciences 16: 1349-1370.

Citation added.

Page 4 Line 26: "the total (clear+cloudy) sky analog". It is not very clear to this reader what this means. Could you please rephrase?

Clarification made to manuscript.

Page 9 Line 6: You have chosen to use weekly (7-day) averages. Could you please explain to the reader why you have chosen averages calculated for a period of 7 days? Why not 5 days – or 10 days?

While the choice of seven days was arbitrary as fewer or more days could also have been selected, the averaging was done for computational efficiency in the minimization. This text has been added to the manuscript.

Page 9 Line 24: You examine 5-day averages of Brewer measurements. Could you please justify the use of 5-day averages? Why not 7-day averages here?

Again arbitary. It was also partially limited by the number of coincident Brewer measurements, made under clear sky conditions, which were recorded within ∼2 minutes local time of the analogous model data produced for the July-August 2015 period. This has been added in the manuscript.

Page 9 Line 23. You remind the reader that a boxcar averaging window was used for the OMI composite TOA spectrum and point out that the slit function of a Brewer spectrophotometer is trapezoid-shaped. The Brewer spectra can be purged from the effects of the slit function by performing a deconvolution. Could you please briefly discuss on how much the different schemes, averaging with a boxcar window vs. convolution with a triangular slit function, may be estimated to affect to the spectra.

Before proceeding, it is necessary to point out that the text should have referred to an approximately triangular-shaped slit function (not trapezoid-shaped). The text has been corrected. Deconvolving the Brewer spectra could have been performed for the model v. instrument comparison, but would have been an involved process. This not only in considering the Brewer slit function, but also in accounting for the spectral variability present in the TOA solar spectrum in the process. An alternative would have been to apply a triangle-shape averaging function to the TOA spectrum for the simulations instead of the boxcar approach. This would have shown the difference in implications of the two averaging approaches. Notable disparities are visually observed at relative extrema points in some of the plots seen in Fig. 4, suggesting that the differences of averaging functions may play a notable role in these disparities. We preferred not doing this as the overall consistency in spectral shape between the simulated and measured data is sufficient for this work. Note that the text of that section has also been revised due to improvements/corrections in the calculations as pointed out at the beginning of this document.

In addition, it was desired to focus on the re-processing and regenerating the figures and updating the text following the corrections identified above in the preamble (and the adjustment of the applied erythemal action spectrum indicated above).

Page 16 Line 2: "The simulated broadbands". I think it should be "The simulated broadband irradiances". There are some other instances in the body text with the same kind of formulation where the actual physical quantity is missing, like on Page 16 Line 28: "all sky broadbands" or Page 14 Line 8: "GEM broadbands". Please add the name of the physical quantity wherever it is currently missing.

Corrections made. 'All sky' has also been changed to 'all-sky' for consistency with use of 'clear-sky'.

Page 17 Line 33: What is a "spectral broadband"? Please explain the term.

The coarse spectral resolution GEM irradiance broadbands. The explanation has been added to the manuscript.

Page 18 (Conclusions). The reader would be extremely interested in any estimate on how much your approach would save computer time as compared to the current operational UV index forecasting. Would you please be able to give an estimate on that?

Neglecting the limitation of the current operational UV index forecasting in providing good UV Index values essentially only over parts of Canada and the northern U.S. (the new setup allows for global coverage at whatever model resolution is available), there are two phases to consider. One is providing the ozone field and or the GEM weather variable or irradiance fields. The second is the calculation of the UV Index itself from the ozone and GEM model output.

The first phase of the two methods are quite different. The operational approach first requires the calculation of total column ozone from weather fields over a predetermined northern hemisphere grid. On the other hand the setup in this paper requires that ozone field assimilation and forecasting be performed first. This by itself would be much more computationally expensive. On the other hand, the ozone assimilation and forecasting process is also intended to benefit other applications. The ozone field forecast is then provided, instead of an ozone climatology, to the model radiation code applied for weather forecasting and so does not add any cost.

For this second phase, it is not believed there would be much or any computational advantage. The calculation for this new setup requires the scaling of the GEM UV surface broadband total irradiances and their application in the integration or linear interpolation approaches. Considering the equations involved, the linear interpolation approach might be a faster and the integration approach could be similar if not a bit slower. The integration was still made to be computationally quite efficient. A few repeat UV Index calculation runs of $\sim$11500 points for each case required, on average, $\sim$0.08 seconds for the operational case and the integration approach and about $\sim$0.07 seconds for the linear interpolation (assuming the units are correct for the conversion of processor clock counts to seconds), with some calculations performed being common to all three cases. This phase of the calculations does not imply any significant time as compared to model forecasting (and estimating the total column ozone for the operational case.